# Healthcare Insurance Fraud Detection via Continual Fiedler Vector Graph Model

**Yehan Zhang**[1]    **Huaidong Zhang**[1,*]    **Xuandi Luo**[1]    **Shengfeng He**[2]

[1]South China University of Technology, [2]Singapore Management University

{yehanzhang,ftxuandi.luo}@mail.scut.edu.cn, huaidongz@scut.edu.cn, shengfenghe@smu.edu.sg

## Abstract

Healthcare insurance fraud detection presents unique machine learning challenges: labeled data are scarce due to delayed verification processes, and fraudulent behaviors evolve rapidly, often manifesting in complex, graph-structured interactions. Existing methods struggle in such settings. Pretraining routines typically overlook structural anomalies under limited supervision, while online models often fail to adapt to changing fraud patterns without labeled updates. To address these issues, we propose the Continual Fiedler Vector Graph model (ConFVG), a fraud detection framework designed for label-scarce and non-stationary environments. The framework comprises two key components. To mitigate label scarcity, we develop a Fiedler Vector-guided graph autoencoder that leverages spectral graph properties to learn structure-aware node representations. The Fiedler Vector, derived from the second smallest eigenvalue of the graph Laplacian, captures global topological signals such as community boundaries and connectivity bottlenecks, which are patterns frequently associated with collusive fraud. This enables the model to identify structurally anomalous nodes without relying on labels. To handle evolving graph streams, we propose a Subgraph Attention Fusion (SAF) module that constructs neighborhood subgraphs and applies attention-based reweighting to emphasize emerging high-risk structures. This design allows the model to adapt to new fraud patterns in real time. A Mean Teacher mechanism further stabilizes online updates and prevents forgetting of previously acquired knowledge. Experiments on real-world medical fraud datasets demonstrate that the Continual Fiedler Vector Graph model outperforms state-of-the-art baselines in both low-label and distribution-shift scenarios, offering a scalable and structure-sensitive solution for real-time fraud detection. Codes are available at https://github.com/yhzhang1309/ConFVG.

## 1 Introduction

Healthcare insurance fraud has become a critical issue in social healthcare systems, with increasingly sophisticated and large-scale fraudulent schemes causing substantial financial losses. According to the 2024 Medicaid Fraud Control Units Annual Report (U.S. Department of Health and Human Services, Office of Inspector General, 2024), Federal Medicare fraud in the U.S. reached approximately $61 billion, accounting for around 7% of total federal healthcare spending. These fraudulent claims not only undermine patients' access to rightful medical services but also strain healthcare budgets and obstruct the path toward universal healthcare. This pressing societal issue calls for more intelligent and scalable fraud detection solutions.

Machine learning has shown promise in identifying fraud patterns by leveraging the complex relationships between entities such as patients, providers, and claims. In particular, graph-based methods have demonstrated strong performance in capturing structural dependencies within healthcare data. For instance, Dou et al. (2020) introduced a graph neural network (GNN) model with selective neighbor aggregation for offline fraud detection. Building on this, Sadreddin and Sadaoui (2022)

---

*Corresponding Author.

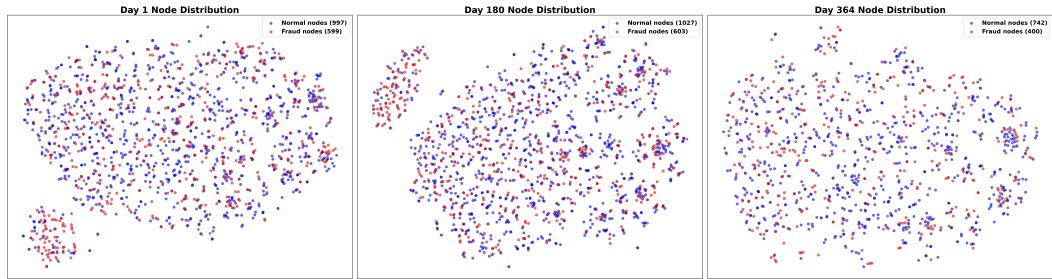

Figure 1: The visualization of fraud pattern evolution in the medical insurance fraud dataset. We use t-SNE to visualize the node distribution of features. The red node denotes fraudulent nodes, whereas blue denotes normal nodes

developed an adaptive learning method that combines transfer and incremental learning. In the online setting, Zhang et al. (2024) applied contrastive learning with parameter-level continual updates to mitigate knowledge forgetting.

Despite these advancements, existing methods remain limited in real-world fraud detection scenarios due to two major challenges: 1) *Label scarcity during model pretraining.* In practice, labeled fraud cases are rare due to the high cost and time required for manual verification (Palacio, 2019), especially in health care systems in some cases where only 0.062% are identified as fraud (Bauder et al., 2018). Most prior methods assume access to fully labeled data during pretraining or rely on supervised objectives, which suffer a decline in performance when labeled supervision is limited. Although some self-supervised approaches employ the autoencoder to catch inner semantic representations(Hou et al., 2022), they often fail to emphasize structurally meaningful fraud patterns such as collusive groups or community anomalies.

2) *Lack of adaptability in non-stationary graph streams.* Fraud patterns are not static as shown in Figure 1. They evolve over time, often emerging in new relational structures. Traditional static fraud detection cannot catch the progressive evolution of fraudulent behaviors, causing performance degradation over time. Meanwhile, labeled data are rarely available in online testing (Pereira and Silveira, 2019), which makes supervised updates impractical. Models based on static presentations or ground-truth label update strategies struggle to detect new fraudulent patterns.

To address these challenges, we propose the *Continual Fiedler Vector Graph model (ConFVG)*, a fraud detection model designed for label-scarce and non-stationary environments. ConFVG (shown in Figure 2) consists of two core modules. 1) To handle label scarcity, we develop a *Fiedler Vector-guided graph autoencoder* that incorporates spectral information from the second smallest eigenvector of the graph Laplacian (Fiedler, 1973). This vector captures global structural properties, such as community boundaries and connectivity bottlenecks, which are often associated with collusive fraud. By integrating these signals into the autoencoder, the model learns structure-aware node representations that highlight fraud-relevant patterns without requiring labeled data. 2) To adapt to evolving fraud behaviors, we introduce a *Subgraph Attention Fusion (SAF) module* that dynamically augments and reweights local subgraphs based on attention. This enables the model to prioritize emerging high-risk structures, such as unusually dense clusters or temporally synchronized activity. A Mean Teacher architecture is used to stabilize updates and prevent catastrophic forgetting during online training. Experiments on real-world datasets, including but not limited to insurance fraud, demonstrate that the ConFVG consistently outperforms state-of-the-art baselines in both low-label and distribution-shift scenarios, providing a scalable and structure-aware solution for real-time fraud detection and beyond.

Our main contributions are as follows:

- We propose the Continual Fiedler Vector Graph model (ConFVG) for fraud detection under label-scarce and non-stationary conditions, integrating spectral self-supervision with adaptive online learning.

- We develop a Fiedler Vector-guided graph autoencoder that leverages global topological signals to learn structure-aware representations in the absence of labels.

- We introduce a Subgraph Attention Fusion (SAF) module for unsupervised online learning, enabling dynamic risk representation and continual adaptation to evolving fraud patterns.

## 2 RELATED WORK

**Fraud Detection.** Fraud detection has become an active area of research across various domains, including credit card transactions, insurance claims, and online payments (Cheng et al., 2025). Early systems were largely rule-based or built upon conventional machine learning techniques (Chan et al., 2002; Srivastava et al., 2008; Maes et al., 2002), which proved insufficient for capturing the complex and evolving nature of modern fraud patterns. More recently, deep learning approaches, particularly Graph Neural Networks (GNNs), Long Short-Term Memory (LSTM) networks, and Large Language Models (LLMs), have been widely adopted for fraud detection. Among them, GNNs have emerged as the most effective for modeling the relational structure inherent in fraud data. GNN-based fraud detection methods can be broadly categorized into three scenarios: fully supervised GNNs, semi-supervised GNNs, and online GNNs. Fully supervised models assume access to large-scale labeled datasets, allowing for comprehensive pretraining. For example, CARE-GNN (Dou et al., 2020) and PC-GNN (Liu et al., 2021) use label-aware node selection strategies to improve fraud detection performance. Semi-supervised GNNs operate under the more realistic assumption of label scarcity. They exploit graph structure through self-supervision and pseudo-labeling. For instance, SemiGNN (Wang et al., 2019) incorporates hierarchical attention, and GTAN (Xiang et al., 2023) uses gated temporal attention to improve fraud representation learning. Online GNNs are designed for dynamic environments where fraud patterns evolve over time, requiring continual updates post-pretraining. POCL (Zhang et al., 2024), for example, adjusts parameters based on their contribution to previous tasks.

Although prior studies have explored either label-efficient pretraining or adaptive fraud detection in evolving environments, these directions have largely been treated in isolation. In real-world medical insurance streams, however, extreme label scarcity and shifting fraud patterns often occur simultaneously. What remains missing is a unified approach that can both generalize from limited supervision and adapt continuously to unseen behaviors. Our work addresses this gap by jointly tackling the structural complexity of fraud graphs and the temporal dynamics of fraud evolution, providing a more realistic and robust solution for fraud detection under deployment constraints.

**Online Learning.** Online learning addresses the challenge of training models on continuously arriving data while mitigating catastrophic forgetting (Hoi et al., 2021). Existing approaches can be broadly classified into three categories: parameter regularization, data rehearsal, and dynamic architecture. Parameter regularization methods constrain model updates to preserve previously learned knowledge. Techniques such as EWC (Li et al., 2020) and EVCL (Batra and Clark, 2024) apply regularization penalties to sensitive parameters, balancing plasticity and stability across tasks. Data rehearsal strategies, including iCaRL (Rebuffi et al., 2017) and GEM (Lopez-Paz and Ranzato, 2017), maintain a buffer of past samples or prototypes to reduce forgetting. Dynamic network approaches adapt the model structure itself to accommodate new knowledge. Notable examples include DEN (Yoon et al., 2017) and BC-DEN (Yang et al., 2022), which expand network capacity based on task complexity. Several methods have applied these ideas to GNNs. For instance, ContinualGNN (Wang et al., 2020) and SGNN-GR (Wang et al., 2022b) combine parameter regularization with data replay for online fraud detection. However, most of these approaches assume access to ground-truth labels during online updates, which is rarely the case in real-world fraud detection scenarios. To address this, we adopt a Mean Teacher framework in combination with our SAF module, enabling unsupervised, structure-aware updates that integrate new information while preserving previously learned representations.

## 3 PROPOSED METHOD

### 3.1 PROBLEM DEFINITION

In online fraud detection scenario, we generate graph series $\mathcal{G} = (\mathcal{G}_0, \mathcal{G}_1, \mathcal{G}_2, \ldots, \mathcal{G}_{|\mathcal{G}|})$ from raw dataset where each graph $\mathcal{G}_i$ represents the graph composed of claims in time periods. In medical insurance claim graph $\mathcal{G}(\mathcal{X}, \mathcal{E}, \mathcal{Y})$, each node $\mathcal{V}_i$ has its features represented by $\mathcal{F}_i \in \mathbb{R}^{1 \times d}$, constituting the node feature matrix $\mathcal{X} \in \mathbb{R}^{n \times d}$. $\mathcal{E} \in \mathbb{R}^{n \times n}$ denotes the edges between claim nodes, where claims have the same medical provider or beneficiary get connected. Node label $\mathcal{Y} \in \mathbb{R}^{n \times 1}$ denotes the label of each node: fraud (1), not fraud (0), or none (-1).

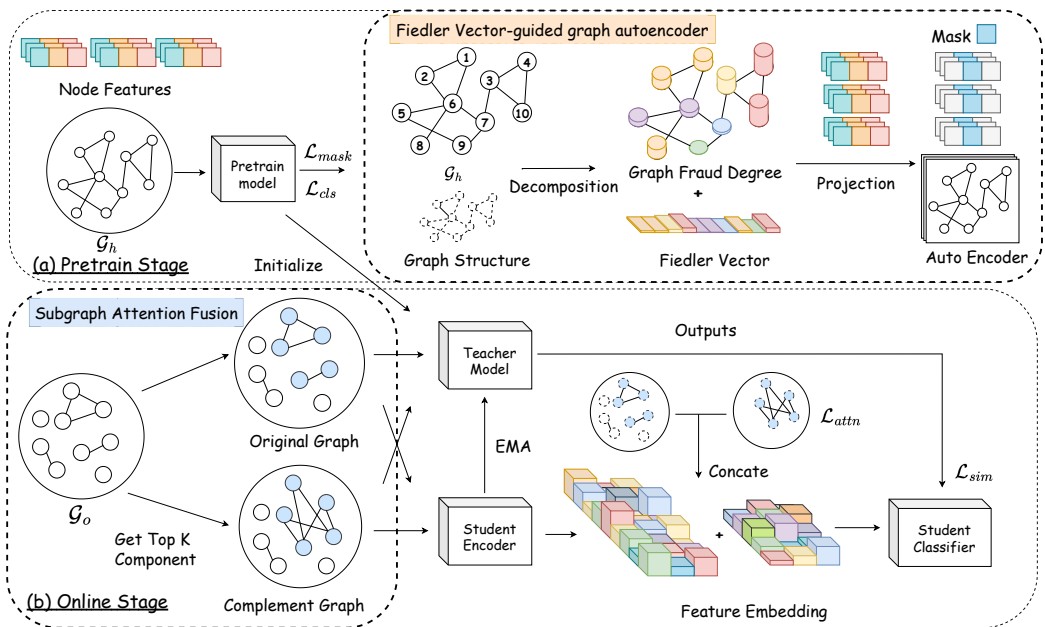

Figure 2: The illustration of our Continual Fiedler Vector Graph model (ConFVG). (a) Pretrain Stage employs Fiedler Vector-guided graph autoencoder to learn structure-aware node representations. (b) Online learning stage utilizes Subgraph Attention Fusion (SAF) module to construct neighborhood subgraphs and apply attention-based reweighting to emphasize high-risk structures.

To demonstrate the continual learning problem, we divide the whole dataset into the historical dataset $\mathcal{G}_h = (\mathcal{G}_h^0, \mathcal{G}_h^1, \mathcal{G}_h^2, \ldots)$ and the online dataset $\mathcal{G}_o = (\mathcal{G}_o^0, \mathcal{G}_o^1, \mathcal{G}_o^2, \ldots)$. In the pretrain stage, the model can only access the historical dataset $\mathcal{G}_h$ to train the parameter $\theta_{history}$. While in the online learning stage, tasks come as a long sequence $\mathcal{G}_o$ and the model can only access the current task $\mathcal{G}_o^i$ without access to the previous dataset. We update the parameters $\theta_i$ of the model after the new incoming task $\mathcal{G}_o^i$ to adapt to the ever-changing environment.

However, in real-world scenarios, historical datasets are scarce, and labeled samples only occupy a small portion (Lebichot et al., 2016), the model accesses only unlabeled samples in the online learning phase. To simplify the problems above, we design a new challenging scenario with a partially labeled dataset in the pretrain stage and an unlabeled dataset in the online stage, as detailed in the experiment.

## 3.2 FIEDLER VECTOR-GUIDED GRAPH AUTOENCODER

In semi-supervised scenarios, the autoencoder is commonly used for providing additional information to the supervised learning process. Given the model encoder $E$ and decoder $D$, our goal is to regenerate node features $\mathcal{X}$ from the feature embeddings $E(\mathcal{G})$ to force the model encoder to learn the deeper feature representation of the graph. For the purpose of node classification, we use feature mask (Li et al., 2023) to randomly mask a part of the node features and reconstruct the masked features by comparing them to the real ones.

For the vanilla random masking method (Hou et al., 2022), given a graph $\mathcal{G}(\mathcal{X}, \mathcal{E}, \mathcal{Y})$ and mask ratio $r$, we obtain the masked node features $\mathcal{X}_{mask}$ by applying the mask matrix $\mathbf{M} \in \{0,1\}^{n \times d}$ to $\mathcal{X}$, where $\mathbf{M}_{if} \sim \text{Bernoulli}(1-r), i \in \{1, 2, \ldots, n\}, f \in \{1, 2, \ldots, d\}$. After obtaining the reconstruction features $\hat{\mathcal{X}} = D(E(\mathcal{G}(\mathcal{X}_{mask}, \mathcal{E}, \mathcal{Y})))$, we use a mean squared error (MSE) loss $\mathcal{L}_{mask}$ to pull closer $\mathcal{X}$ and $\hat{\mathcal{X}}$ as follows

$$\mathcal{L}_{\text{mask}} = \frac{1}{|\mathcal{I}|} \sum_{i=1}^{n} \sum_{f=1}^{d} (1 - \mathbf{M}_{if}) \cdot (\mathcal{X}_{if} - \hat{\mathcal{X}}_{if})^2, \text{ where } \mathcal{I} = \{(i, f) \mid \mathbf{M}_{if} = 0\}. \tag{1}$$

However, the vanilla random mask is not solid and may mask some of the fruitless features or fail to mask important features (Liu et al., 2024), which may cause information loss and semantic

irrelevance. To address the limitations of the above masking techniques, we incorporate the global structural information with node features to take into account the fraud importance to the masking strategies.

In the context of graph representation for fraud detection, fraudulent nodes often exhibit significant anomalous connectivity patterns (Pourhabibi et al., 2020), such as isolated nodes or abnormally dense local subgraphs. These structural deviations disrupt the global homophily of the graph, leading to sharp increases in feature or label discrepancies among node neighborhoods. From the perspective of spectral graph theory, such anomalous connectivity can be interpreted as 'non-smooth' signals: normal nodes tend to maintain smooth representations within communities, while fraudulent nodes introduce strong 'non-smooth' values and high local gradients. This phenomenon is illustrated in Appendix A.1.

To capture these non-smooth nodes and find out the fraud features, given a graph $\mathcal{G}(\mathcal{X}, \mathcal{E}, \mathcal{Y})$, we calculate its degree matrix $\mathbf{D}$, adjacency matrix $\mathbf{A}$, and laplacian matrix $\mathbf{L} = \mathbf{D} - \mathbf{A}$ with the edge set $\mathcal{E}$. To better quantify the smoothness of the node, we denote each node $i$ has its smoothness value $q_i$, which constitutes the smoothness vector $\mathbf{q} \in \mathbb{R}^{n \times 1}$. We defined the global smoothness $Graph_{smooth}$ as follows

$$Graph_{smooth} = \sum_{(i,j) \in \mathcal{E}} (q_i - q_j)^2. \tag{2}$$

To better optimize the smoothness of each node, we hypothesize that the connected nodes share similar smoothness values, while the disconnected nodes have dissimilar values. Therefore, we aim to minimize $Graph_{smooth}$ as follows

$$\min \sum_{(i,j) \in \mathcal{E}} (q_i - q_j)^2 = \min \mathbf{q}^T \mathbf{L} \mathbf{q} = \min \sum_{i=1}^{n} \lambda_i z_i^2, \tag{3}$$

where $\lambda_i$ denotes the eigenvalue of $\mathbf{L}$, and $z_i$ is the projection of $\mathbf{q}$ on eigenvector $i$. To ensure the Eq. 3 is minimized and meaningful, we make $z_1 = 0, z_2 = 1, z_3 = \cdots = z_n = 0$. So $\mathbf{q}^T \mathbf{L} \mathbf{q} = \lambda_2$, where $\mathbf{q}$ is the eigenvector $u_2$ corresponding to $\lambda_2$, as the Fiedler vector $v_f = u_2$ indicating the fraudulent probability of each node.

In the medical insurance fraud scenario, graphs are not always connected and may even have multiple connected components. Because of that, the Laplacian spectral decomposition in Eq. 3 degenerates, and the Fiedler vector loses its ability to capture anomalies. To address this problem, we add a fully connected perturbation to the original matrix, weakly connecting the multiple connected components into a single connected graph as follows

$$\mathbf{A}' = \mathbf{A} + \epsilon \cdot (\mathbf{J} - \mathbf{I}), \tag{4}$$

where $\mathbf{A}'$ is the perturbed adjacency matrix, $\epsilon$ is the perturbation parameter, $\mathbf{J}$ is a matrix of all ones, and $\mathbf{I}$ is the identity matrix. The bound of perturbation is discussed in Appendix, A.2.

While the smoothness values of a node solely represent the inner graph structure, we combine that with the semantic features of nodes to show the contribution of each feature. We normalize the node feature matrix and make a linear projection with $\mathbf{x}$ to get the mask probability $s_j$ as follows

$$s_j = \left| \sum_{i=1}^{n} v_{f,i} \cdot X_{norm,ij} \right|, \quad j = 1, 2, \ldots, d. \tag{5}$$

With the mask weight $s$, we can reformulate the mask matrix $\mathbf{M}$ in Eq. 1 with sampling probability equal to $s$, incorporating the fraud importance to the masking strategies.

To summarize, the total loss in the pretrain stage $\mathcal{L}_{pretrain}$ is defined as follows

$$\mathcal{L}_{pretrain} = \mathcal{L}_{cls} + \alpha_{mask} \mathcal{L}_{mask}, \tag{6}$$

where $\alpha_{mask}$ is a hyperparameter of reconstruction loss.

### 3.3 SUBGRAPH ATTENTION FUSION MODULE

#### 3.3.1 SUBGRAPH COMPLEMENT STRATEGY

Traditional graph models, i.e., GraphSAGE(Hamilton and Ying, 2017), GAT(Veličković et al., 2017) use top-k neighbors to transfer information between convolutional layers as follows

$$\mathbf{h}^k_{\mathcal{N}(v)} \leftarrow \text{AGGREGATE}_k\left(\{\mathbf{h}^{k-1}_u, \forall u \in \mathcal{N}(v)\}\right), \tag{7}$$

where $\mathbf{h}$, $\mathcal{N}(v)$ denotes the feature vector and neighbor nodes of node $v$, respectively, and $k$ denotes the $k^{th}$ layer of the model. These methods only focus on the subgraph that is fully connected in the graph, which overlooks the correlation between each connected part. The performance of the model becomes highly dependent on the initial graph connection, resulting in weaker generalization capability. In addition, focusing solely on single connected graph may weaken the model ability to capture similar fraudulent patterns from the entire graph, especially when the environment is dynamic and fraudulent activities are ever-changing.

Therefore, we propose a subgraph attention-fused complement strategy to combine the original graph with its subgraph complement to capture evolving fraudulent behavior. Suppose $\mathcal{G}(\mathcal{X}, \mathcal{V}, \mathcal{Y})$ is a graph with $m$ connected components, denoted as $C_1, C_2, \ldots, C_m$. Each connected component $C_i = (\mathcal{X}_i, \mathcal{V}_i)$ consists of a node feature matrix set $\mathcal{X}_i \subseteq \mathcal{X}$ and a node set $\mathcal{V}_i \subseteq \mathcal{V}$. We can get the graph complement $\mathcal{G}_{comp}$ from the selected connected components as follows

$$\mathcal{G}_{\text{comp}} = \left(\{0, 1, \ldots, |\mathcal{V}| - 1\}, \ \{(\phi(i), \phi(j)) \mid i \neq j, \ i, j \in \mathcal{V}, \ (i, j) \notin \mathcal{E}\}\right) \tag{8}$$

where $\mathcal{V} = \bigcup_{i=1}^{k} C_i$, $\phi()$ is the index projection function, and $k$ denotes the top-k largest components. The value of k is discussed in Appendix, A.3.Then we can obtain the original graph embeddings $\mathbf{z}_{\text{orig}} = E(\mathcal{G})$ and the complement graph embeddings $\mathbf{z}_{\text{comp}} = E(\mathcal{G}_{comp})$. So our attention-fused embedding $z$ is defined as follows

$$z = \sigma(\mathbf{W_2}\text{ReLU}(\mathbf{W_1}[\mathbf{z}_{\text{orig}}; \mathbf{z}_{\text{comp}}] + \mathbf{b}_1) + \mathbf{b}_2), \tag{9}$$

where $\mathbf{W_1}$ and $\mathbf{W_2}$ are parameter matrix for attention-fused, $\mathbf{b}_1$ and $\mathbf{b}_2$ are bias value, and $\sigma$ is the sigmoid active function. To force the model to adaptively capture the fraudulent pattern in the online learning stage, we use the attention loss $\mathcal{L}_{\text{attn}}$ to control its weights on the complement graph as follows,

$$\mathcal{L}_{\text{attn}} = \text{ReLU}\left(\frac{1}{|\mathcal{V} \setminus \mathcal{V}_\mathcal{G}|} \sum_{i \in \mathcal{V} \setminus \mathcal{V}_\mathcal{G}} a_i - \frac{1}{|\mathcal{V}_\mathcal{G}|} \sum_{i \in \mathcal{V}_\mathcal{G}} a_i\right), \tag{10}$$

where $\mathcal{G} = (\mathcal{V}_\mathcal{G}, E_\mathcal{G})$ is the subgraph induced by the selected nodes $\mathcal{V}_\mathcal{G} \subseteq \mathcal{V}$, $\mathcal{V} \setminus \mathcal{V}_\mathcal{G}$ is the set of unselected nodes, $\mathbf{a} \in \mathbb{R}^n$ is the attention weight vector with $a_i$ for node $i$.

#### 3.3.2 MEAN TEACHER STRUCTURE

In the online learning scenario where labels are inaccessible, we utilize the mean-teacher structure (Tarvainen and Valpola, 2017) with $\mathcal{L}_{\text{attn}}$ from Eq. 10 to update its parameter in data streams. We denote the teacher model as $\mathbf{M_t}$, and the student model as $\mathbf{M_s}$, with their parameter $\theta_t$ and $\theta_s$, respectively. The task sequence is denoted by $task = (t_0, t_1, t_2, \ldots)$, where $t_i$ denotes the graph $\mathcal{G}_i$. When a new task $t_i$ comes, the teacher model predicts the outputs, and the student model uses the outputs to update its parameters. We use the KL divergence to restrict the outputs of the teacher model and student model as follows

$$\mathcal{L}_{\text{sim}} = \text{KL}\left(\text{Softmax}(\mathbf{z}_s) \,\middle\|\, \text{Softmax}(\mathbf{z}_t)\right). \tag{11}$$

So the total loss in the online learning scenario $\mathcal{L}_{\text{online}}$ is

$$\mathcal{L}_{\text{online}} = \mathcal{L}_{\text{sim}} + \alpha_{\text{attn}}\mathcal{L}_{\text{attn}}, \tag{12}$$

where $\alpha_{\text{sim}}$ is a hyperparameter. After $i^{th}$ task, the parameter of the student model gets updated by $\mathcal{L}_{\text{online}}$, while the parameter of the teacher model gets updated using exponential moving average (EMA) as follows

$$\theta_t^{(i)} = \alpha\,\theta_t^{(i-1)} + (1 - \alpha)\,\theta_s^{(i)}. \tag{13}$$

In summary, the pseudocode of the proposed Continual Fiedler Vector Graph Model is presented in Algorithm 1.

---

**Algorithm 1** Continual Fiedler Vector Graph Model

---

1: **Input:** Historical graph $\mathcal{G}_h$, online graph dataset $\mathcal{G}_o$
2: **Output:** Model $\mathbf{M_s}$ with parameter $\theta_s$
3: **Initialize:** Model $\mathbf{M_s}$, decoder $D$
4: Compute graph laplacian matrix $\mathbf{L}$ ,get $v_f$ from decomposition
5: Compute mask weight $s_j = |\sum_{i=1}^{n} v_{f,i} \cdot X_{norm,ij}|$, get mask matrix $\mathbf{M}$
6: Compute $\mathcal{L}_{\text{mask}}$, $\mathcal{L}_{\text{cls}}$
7: Update $\theta_s$ with $\mathcal{L}_{\text{pretrain}} = \mathcal{L}_{\text{cls}} + \alpha_{\text{mask}} \mathcal{L}_{\text{mask}}$, set $\theta_t \leftarrow \theta_s$
8: **for** each online graph $\mathcal{G}_0^i$, $i = 1, 2, 3, \ldots$ **do**
9:     Make top k component graph complement $\mathcal{G}_{comp}$
10:     Fuse attention $z = \sigma(\mathbf{W_2}\text{ReLU}(\mathbf{W_1}[\mathbf{z}_{\text{orig}}; \mathbf{z}_{\text{comp}}] + \mathbf{b}_1) + \mathbf{b}_2)$
11:     Compute $\mathcal{L}_{\text{sim}}$, $\mathcal{L}_{\text{attn}}$
12:     Update $\theta_s$ with $\mathcal{L}_{\text{online}} = \mathcal{L}_{\text{sim}} + \alpha_{\text{attn}} \mathcal{L}_{\text{attn}}$
13:     Update $\theta_t \leftarrow \alpha\theta_t + (1 - \alpha)\theta_s$
14: **end for**

---

## 4 EXPERIMENTS

In scenarios with various proportions of scarce labels, we conduct experiments on the medical health insurance dataset and other real-world fraud datasets. We first introduce the datasets and experimental settings, then validate the model's exceptional performance and generalization capability under real-world scenarios. Subsequently, we perform ablation studies to demonstrate the importance of each component. Finally, we analyze the role of the autoencoder and subgraph complement strategy through case studies.

### 4.1 DATASET AND EXPERIMENT SETUP

**Dataset and metrics.** To evaluate the performance of our model on real-world datasets, we utilized a large-scale medical insurance dataset (Ma et al., 2023). This dataset comprises over 100,000 unique beneficiaries with a total of 517,737 distinct claims. All samples in the dataset are labeled and timestamped, facilitating temporal grouping and validation. In the experiments, we select one year of the data and construct daily medical diagnosis graphs. The initial 15 days were designated as the historical dataset, while the remaining served as the online learning dataset. To better simulate real-world scenarios, we randomly retain labels in the historical dataset with probabilities of 1%, 10%, while removing all labels from the online learning dataset. To demonstrate the generalizability of our model across different scenarios, we conducted additional experiments using two widely used datasets: Amazon (McAuley and Leskovec, 2013) and YelpChi (Rayana and Akoglu, 2015). As the evaluation metric, we use the average accuracy, F1 score, and AUC to measure the model's performance in online learning scenarios.

**Baseline.** We compare our method with the state-of-the-art (SOTA) GNN models in fraud detection, including well-established baselines: CAREGNN(Dou et al., 2020), PCGNN (Liu et al., 2021), SAD(Tian et al., 2023), GTAN(Xiang et al., 2023), GAD (Chen et al., 2024), ContinuesGNN(Wang et al., 2020), FGN(Wang et al., 2022a), POCL(Zhang et al., 2024). CAREGNN and PCGNN are traditional offline models, while SAD, GTAN, and GAD are designed for the semi-supervised scenario. FGN and POCL are online models with parameter updates. In light of previous work not applicable to the unsupervised learning scenario, we utilize their pre-trained model to complete the subsequent online learning. To ensure the fairness of comparison, we make extra experiments on a traditional online learning scenario where labels are abundant, denoted as 100%* label rate.

### 4.2 EXPERIMENTAL RESULTS

We conduct extensive experiments to compare our model against SOTA baseline methods. As shown in Table 1, our model demonstrates superior performance, achieving the highest AUC and F1-score across real-world scenarios with different label ratios. With a 10% label rate, a fully supervised

Table 1: Performance comparison of different models on medical insurance dataset at different label rates. 100% * label rate denotes the traditional online scenario.

| Model Type | Model | 1% label rate | | 10% label rate | | 100% * label rate | |
|---|---|---|---|---|---|---|---|
| | | AUC | F1 Score | AUC | F1 Score | AUC | F1 Score |
| Traditional Offline | CAREGNN | 62.08 ± 1.31 | 40.58 ± 3.05 | 67.35 ± 1.20 | 49.61 ± 1.50 | 75.12 ± 0.78 | 54.23 ± 0.24 |
| | PCGNN | 63.75 ± 1.72 | 50.16 ± 2.14 | 69.38 ± 0.55 | 54.25 ± 0.91 | 78.11 ± 0.33 | 60.10 ± 0.55 |
| Semi-Supervised Offline | SAD | 72.58 ± 1.29 | 55.40 ± 1.95 | 76.12 ± 1.23 | 62.30 ± 2.24 | 78.56 ± 0.21 | 62.04 ± 0.41 |
| | GTAN | 72.43 ± 1.30 | 57.33 ± 1.71 | 75.60 ± 1.12 | 61.38 ± 2.57 | 76.54 ± 0.53 | 61.29 ± 0.67 |
| | GAD | 73.29 ± 1.11 | 56.81 ± 2.19 | 76.54 ± 0.83 | 61.73 ± 1.88 | 77.56 ± 0.18 | 62.35 ± 0.28 |
| Traditional Online | ContinuesGNN | 63.14 ± 1.07 | 49.22 ± 2.83 | 69.52 ± 1.97 | 53.26 ± 1.36 | 78.60 ± 0.10 | 57.32 ± 0.22 |
| | FGN | 62.83 ± 1.21 | 43.30 ± 3.21 | 65.71 ± 1.01 | 50.11 ± 1.92 | 73.91 ± 0.25 | 56.42 ± 0.51 |
| | POCL | 70.64 ± 1.34 | 52.45 ± 2.57 | 74.76 ± 1.12 | 60.31 ± 1.88 | 80.32 ± 0.21 | **63.56 ± 0.28** |
| Proposed | Ours | **76.13 ± 1.05** | **62.24 ± 2.26** | **80.48 ± 0.86** | **64.48 ± 1.24** | **80.61 ± 0.08** | 63.24 ± 0.16 |

Table 2: Performance comparison of models across different real-world datasets at 10% label rate.

| Model Type | Model | Medical | | Yelpchi | | Amazon | |
|---|---|---|---|---|---|---|---|
| | | AUC | F1 Score | AUC | F1 Score | AUC | F1 Score |
| Traditional Offline | CAREGNN | 67.35 ± 1.20 | 49.61 ± 1.50 | 71.12 ± 1.83 | 61.05 ± 2.12 | 87.31 ± 0.06 | 84.24 ± 0.26 |
| | PCGNN | 69.38 ± 0.55 | 54.25 ± 0.91 | 73.45 ± 1.17 | 61.21 ± 1.60 | 88.58 ± 0.12 | 86.12 ± 0.35 |
| Semi-Supervised Offline | SAD | 76.12 ± 1.23 | 62.30 ± 2.24 | 73.17 ± 0.59 | 61.01 ± 2.23 | 87.53 ± 0.62 | 84.16 ± 0.99 |
| | GTAN | 75.60 ± 1.00 | 61.38 ± 2.57 | 74.74 ± 1.01 | 62.22 ± 2.01 | 88.67 ± 0.20 | 83.16 ± 0.77 |
| | GAD | 76.54 ± 0.83 | 61.73 ± 1.70 | 75.22 ± 0.96 | 62.61 ± 1.93 | 89.56 ± 0.18 | 85.05 ± 0.31 |
| Traditional Online | ContinuesGNN | 69.52 ± 1.97 | 53.26 ± 1.36 | 71.62 ± 1.68 | 60.95 ± 2.05 | 81.28 ± 0.14 | 73.52 ± 0.35 |
| | FGN | 65.71 ± 1.01 | 50.11 ± 1.92 | 70.37 ± 2.01 | 56.41 ± 2.57 | 81.12 ± 1.14 | 74.56 ± 1.55 |
| | POCL | 74.76 ± 1.12 | 60.31 ± 1.88 | 73.18 ± 0.92 | 61.60 ± 2.11 | 87.57 ± 0.38 | 80.12 ± 1.53 |
| Proposed | Ours | **80.48 ± 0.86** | **64.48 ± 1.24** | **76.85 ± 0.12** | **64.53 ± 1.92** | **91.07 ± 0.04** | **87.32 ± 0.52** |

model like CAREGNN requires full labels for pretraining and updating, which presents a challenge. Although POCL is a fully supervised model, it utilizes the contrastive learning method to enhance its discriminative capability between positive and negative samples, as seen in the pre-training stage, which performs stably when the source dataset is scarce. Semi-supervised models apparently present better performance for their data augmentation mechanics. When the label rate drops to 1%, traditional methods perform much worse with infeasible supervised information. And semi-supervised methods drop less. In contrast, our model shows excellent performance when the label rate drops from 10% to 1% with minimal degradation, which highlights the superiority of our approach.

Besides, we also make an experiment on a traditional scenario denoted as 100% * label rate, where the pretrain and online datasets are fully labeled. We can see that the traditional offline model is the worst, while the semi-supervised model performs a little better at catching the inner pattern for fraud activities, which is a good backbone for the upcoming task. But they cannot update themselves to adapt to the dynamic environment. Traditional online models update themselves in the online learning stage and perform better. Interestingly, our model still presents its SOTA performance compared to the traditional online model (despite the minimal difference in F1 score). We think that in the historical dataset, our graph autoencoder has already learned nearly complete fraud patterns. Furthermore, the subgraph completion strategy in the online learning phase accurately captures and translates domain shifts in fraudulent behavior, which boosts our model performance in a traditional scenario.

To better visualize the performance of the methods above during the online learning stage, we employ the monthly average accuracy to measure the performance. As shown in Figure 3, in the 10% labeled data scenario, our model and semi-supervised model achieve a relatively high initial accuracy. During the online learning process, our model stays solid in accuracy by adapting to fraudulent domain shifts. Other models have a certain degree of accuracy reduction, especially for traditional ones. With 1% labeled data, our model still demonstrates a notably high initial accuracy and better adapts to the environment with higher accuracy, while other models exhibit larger fluctuations in accuracy. Most of the methods suffer from a continual decrease over time. To further validate our method on real-world scenarios, we conduct extensive experiments on Yelpchi and Amazon datasets. As shown in Table 2, our model greatly outperforms other methods.

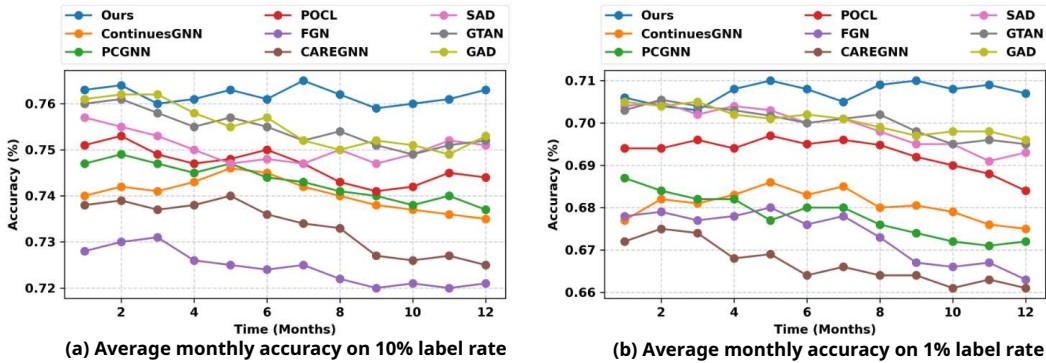

Figure 3: Performance comparison of different fraud detection methods with different label rates on average monthly accuracy and average accuracy decline rate for online learning within a year in the medical fraud dataset.

Table 3: Ablation study on medical insurance fraud dataset with different component combinations.

| Autoencoder | Graph Complement | Mean-Teacher | Medical Dataset | | |
|---|---|---|---|---|---|
| | | | AUC | F1-Score | Accuracy |
| × | × | × | 67.21 ± 2.41 | 39.13 ± 5.62 | 63.43 ± 2.12 |
| × | × | ✓ | 67.43 ± 2.12 | 40.52 ± 4.03 | 63.21 ± 2.01 |
| × | ✓ | × | 68.35 ± 2.06 | 46.41 ± 3.47 | 64.55 ± 2.31 |
| ✓ | × | × | 76.13 ± 0.88 | 61.56 ± 1.10 | 74.29 ± 1.05 |
| ✓ | ✓ | × | 78.21 ± 1.23 | 63.11 ± 1.52 | 74.12 ± 1.11 |
| × | ✓ | ✓ | 72.08 ± 1.78 | 60.24 ± 2.11 | 66.56 ± 1.75 |
| ✓ | × | ✓ | 77.35 ± 1.54 | 64.25 ± 2.53 | 73.81 ± 1.92 |
| ✓ | ✓ | ✓ | **80.48 ± 0.86** | **64.48 ± 1.24** | **76.45 ± 0.62** |

### 4.3 ABLATION STUDY

#### 4.3.1 COMPONENT ANALYSIS

To evaluate the role of each component in the medical insurance fraud model, we conduct a series of ablation studies on the label rate 10% scenario.

As shown in Table 3, we observe that the model performs weakly with only one component for a poor pretrain model or online parameter update. Combining the graph autoencoder with the mean-teacher framework yields better results, yet the addition of the graph completion mechanism is still necessary to enhance the learning of fraudulent representation. Without the graph autoencoder, the model has suboptimal performance due to the weak backbone. Finally, the integration of all three components shows the best performance. This whole model not only strengthens the generalization of fraud patterns from the pretrain model but also leverages the advantages of the graph completion strategy during online learning, enabling effective dynamic adaptation to evolving environments.

#### 4.3.2 ANALYSIS ON DIFFERENT MASK STRATEGY

To prove our extraordinary performance of mask strategy, we compare our proposed Fiedler Vector-based feature masking approach with existing advanced methods, including random masking (Hou et al., 2022), edge-based masking (Tan et al., 2023), and subgraph masking (Jiao et al., 2024), as well as VGAE (Kipf and Welling, 2016). As shown in Table 4, the Fiedler vector-based graph autoencoder selectively reconstructs the core features to capture deeper fraudulent representation, outperforming the other baseline methods.

### 5 CONCLUSION AND FUTURE WORK

In this paper, we introduce ConFVG, which integrates a Fiedler Vector-guided graph encoder and a Subgraph Attention Fusion module to address the challenges of pre-training label scarcity and

Table 4: Performance comparison of different mask strategies on medical insurance fraud dataset with label ratio of 10%.

| Mask Strategies | AUC | F1-score | Accuracy |
|---|---|---|---|
| Random Mask(Hou et al., 2022) | 74.14 ± 0.68 | 59.23 ± 1.28 | 68.46 ± 1.50 |
| Edge Mask(Tan et al., 2023) | 73.25 ± 1.23 | 58.36 ± 2.13 | 67.15 ± 1.31 |
| Subgraph Mask(Jiao et al., 2024) | 75.24 ± 0.93 | 56.15 ± 1.23 | 69.59 ± 1.78 |
| VGAE(Kipf and Welling, 2016) | 77.87 ± 1.86 | 63.24 ± 1.02 | 74.25 ± 0.24 |
| Fiedler-vector Mask | **80.48 ± 0.86** | **64.48 ± 1.24** | **76.45 ± 0.62** |

unlabeled online learning in real-world scenarios. Our model leverages the graph encoder to learn robust representations of fraudulent patterns, while incorporating a subgraph complement strategy to enrich fraud representations during the online learning phase. We conduct extensive experiments on multiple real-world datasets under two semi-supervised settings. The results demonstrate that our model significantly outperforms existing methods in semi-supervised scenarios while remaining competitive in traditional settings. In the future, we aim to extend our approach to other graph anomaly detection tasks, such as network security intrusion detection and abnormal behavior identification in social networks. We also plan to explore the integration of multimodal graph data to further enhance feature extraction and graph representation learning capabilities.

## ACKNOWLEDGMENTS

This work was supported by the National Natural Science Foundation of China (Grant No. 62302170), Guangdong Basic and Applied Basic Research Foundation (Grant No. 2024A1515010187), Guangdong Natural Science Funds for Distinguished Young Scholars (Grant No. 2023B1515020097), the National Research Foundation, Singapore under its AI Singapore Programme (AISG Awards No. AISG3-GV-2023-011 and AISG4-TC-2025-018-SGKR), and the Lee Kong Chian Fellowships.

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

# A APPENDIX

## A.1 STUDY ON FIEDLER VECTOR

To better explain the relationship between fraudulent nodes and Fiedler vector, we choose four slices medical insurance claim graph and visualize the distribution of the Fiedler value (smoothness value ) in each node. The results are shown in Figure 4.

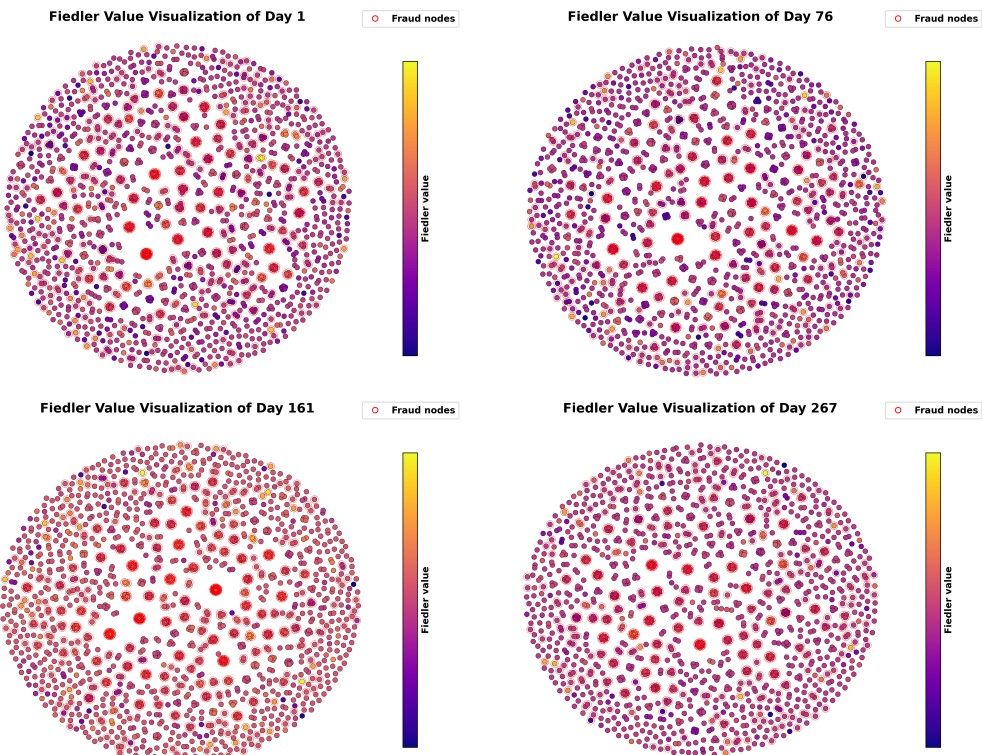

Figure 4: The visualization of Fiedler value in four medical insurance claim graphs. We use spring layout to visualize the node distribution, where connected nodes are put closer while disconnected nodes are put further. The fraudulent nodes are circled in red.

From Figure 4, we can easily conclude: 1) Connected nodes share similar Fiedler value (smoothness value) in analogous colors. 2) Fraudulent nodes always exhibit isolated nodes or dense local subgraphs like little clusters in the center of the graph, which are identified by the Fiedler vector with a higher Fiedler value. Additionally, we conduct a statistical analysis on the same dataset. The results shown in Table 5 also demonstrate the strong guidance of the Fiedler vector.

## A.2 BOUND OF PERTURBATION

In our fraud detection framework, the difference between the original null space $U$ and the perturbed subspace $U'$ arises from added noise via the perturbation $E = \epsilon(nI - J)$. To assess whether the perturbed Fiedler vector $v'_2$ retains indicative significance for anomaly detection, we need to evaluate its closeness to the original null space $U$, measured by the projection error $e_{proj} = \|v'_2 -$

Table 5: Statistics of nodes in the medical insurance dataset across days. We use ratio = mean fraud node Fiedler value / mean normal node Fiedler value to measure the detection ability.

| Day | All Nodes | Fraud Nodes | Normal Nodes | Edges | Ratio (Fraud / Normal) |
|-----|-----------|-------------|--------------|-------|------------------------|
| 1 | 1596 | 599 | 997 | 3616 | 1.864 |
| 50 | 1608 | 616 | 992 | 3996 | 2.017 |
| 100 | 1599 | 602 | 997 | 3664 | 1.857 |
| 150 | 1608 | 586 | 1022 | 4014 | 2.141 |
| 200 | 1555 | 577 | 978 | 3292 | 1.922 |
| 250 | 1467 | 602 | 865 | 3218 | 1.837 |
| 300 | 1440 | 573 | 867 | 2970 | 2.032 |
| All | 554407 | 211270 | 343137 | 1255698 | 1.924 |

$P_U v_2'\|_2$. A smaller error $e_{proj}$ indicates closer alignment with $U$, preserving the soft indicator function structure.

To measure the error, we make an orthogonal decomposition of the Fiedler vector $v_2' \in U'$ with respect to the subspace $U$ as follows

$$v_2' = P_U v_2' + (I - P_U)v_2', \tag{14}$$

where $P_U v_2' \in U$ is the orthogonal projection of $v_2'$ onto the null space $U$, and $(I - P_U)v_2' \in U^\perp$ is the component of $v_2'$ in the orthogonal complement of $U$. So the $E_{proj}$ can be derived as follows

$$e_{proj} = \|v_2' - P_U v_2'\|_2 = \|(I - P_U)v_2'\|_2, \tag{15}$$

Given that $v_2'$ is a unit vector, $\|v_2'\|_2 = 1$, it follows that

$$e_{proj} = \|(I - P_U)v_2'\|_2 = \sqrt{1 - \|P_U v_2'\|_2^2}. \tag{16}$$

We denote the angle $\theta(v_2', U)$ as the angle between $v_2'$ and the subspace $U$, given by the angle to the closest vector $u$ in $U$. So we can conclude that

$$\cos\theta(v_2', U) = \max_{u \in U, \|u\|_2 = 1} |\langle v_2', u \rangle|. \tag{17}$$

Apparently, $u$ is equal to the unit vector $\frac{P_U v_2'}{\|P_U v_2'\|_2}$ of the projection of $v_2'$ onto $U$, so we can conclude that

$$\cos\theta(v_2', U) = \langle v_2', \frac{P_U v_2'}{\|P_U v_2'\|_2} \rangle = \|P_U v_2'\|_2. \tag{18}$$

Using the trigonometric identity:

$$\sin\theta(v_2', U) = \sqrt{1 - \cos^2\theta(v_2', U)} = \sqrt{1 - \|P_U v_2'\|_2^2}. \tag{19}$$

Comparing the projection error and the angle:

$$e_{proj} = \sqrt{1 - \|P_U v_2'\|_2^2} = \sin\theta(v_2', U). \tag{20}$$

Table 6: Performance comparison of different $n_c$ on medical insurance fraud dataset on label ratio of 10% with time consumption (seconds).

| $n_c$ | AUC | F1-score | Accuracy | Time |
|-------|-----|----------|----------|------|
| 0.01 | 79.32 ± 1.06 | 63.23 ± 1.73 | 75.22 ± 1.50 | 309.22 ± 2.41 |
| 0.05 | 80.48 ± 0.86 | 64.48 ± 1.24 | 76.45 ± 0.62 | 328.14 ± 2.85 |
| 0.1 | 80.51 ± 0.76 | 64.42 ± 1.10 | 76.22 ± 0.45 | 384.50 ± 2.73 |
| 0.2 | 80.05 ± 0.94 | 64.10 ± 1.02 | 74.25 ± 1.23 | 470.62 ± 3.59 |
| 0.3 | 78.08 ± 1.27 | 62.70 ± 1.56 | 73.45 ± 1.60 | 612.05 ± 5.17 |

By the Davis-Kahan theorem(Davis and Kahan, 1970) in its single-vector form, for any $v \in U'$, the projection error is bounded by the maximum canonical angle between $U$ and $U'$:

$$\sin \theta(v, U) = \|(I - P_U)v\|_2 \leq \|\sin \Theta(U, U')\|_2 \tag{21}$$

Furthermore, the 2-norm of the sine of the angles satisfies:

$$\|\sin \Theta(U, U')\|_2 \leq \|\sin \Theta(U, U')\|_F \leq \frac{\|E\|_2}{\delta}. \tag{22}$$

Thus, for $v_2' \in U'$, we obtain:

$$\|v_2' - P_U v_2'\|_2 = \sin \theta(v_2', U) \leq \|\sin \Theta(U, U')\|_2 \leq \|\sin \Theta(U, U')\|_F \leq \frac{\|E\|_2}{\delta}. \tag{23}$$

As $\|E\|_2 = \epsilon n$ is the spectral norm of the perturbation and $\delta = \min(\lambda_{k+1}(L), \lambda_1(L')) \approx \lambda_{k+1}(L)$, we can derive the upper bound of $e_{proj}$ as follows,

$$E_{proj} = \|v_2' - P_U v_2'\|_2 \leq \frac{\|E\|_2}{\delta} = \frac{\epsilon n}{\lambda_{k+1}(L)}. \tag{24}$$

This bound, approximately 0.04 for medical insurance claim networks (n = 2000, $\epsilon = 10^{-5}$, $\lambda_{k+1}(L) \approx 0.5$), ensures that $v_2'$ remains close to $U$, retaining its ability to capture anomalies via projections like $|X^T v_2'|$.

To prove our assumption above, We test our model with different $\epsilon$ values. The results are shown in Table 7.

Table 7: Performance of the model under different pertubation $\epsilon$ values.

| Perturbation $\epsilon$ | 1e-6 | 1e-5 | 5e-5 | 1e-4 | 1e-3 | 1e-2 |
|---|---|---|---|---|---|---|
| AUC | 78.72 ± 0.53 | 80.48 ± 0.86 | 80.46 ± 0.78 | 80.33 ± 1.02 | 78.30 ± 1.22 | 76.05 ± 1.53 |
| F1-score | 64.02 ± 1.29 | 64.48 ± 1.24 | 64.52 ± 1.33 | 63.67 ± 1.67 | 63.08 ± 1.93 | 61.22 ± 1.86 |

From Table 7, we can see that when $\epsilon$ is very small ($\epsilon < 1e-6$), the pertubation matrix is too weak to overcome the sparsity of the original graph, resulting in an insufficient correction effect on the Fiedler vector. When $\epsilon$ is in the range of $1e-5$ to $1e-4$, the subtle global connections optimally strengthen the connectivity within communities without altering the macroscopic community structure, thereby sharpening the boundaries between communities and better reflecting the true community structure. However, when $\epsilon$ becomes too large ($\epsilon > 1e-3$), the global pertubation matrix overwhelms the original community structure, leading to a decline in performance.

### A.3 STUDY ON TOP K COMPONENT

We conduct a series of experiments to test the best k value for the connected component. Concerning that the number of nodes in each component may be different, we use the indicator $n_c = \frac{node_{sel}}{node_{all}}$ to measure our selection, where $node_{sel}$ denotes the number of nodes in our selected component, $node_{all}$ denotes the number of nodes in the whole graph. Our experiment result is as follows,

As shown in Table 6, the value of $n_c$ achieves a balance between accuracy and efficiency at 0.05. When $n_c$ is lower, the number of selected nodes is too small for the model to effectively utilize attention information during online learning. Conversely, when $n_c$ is higher, excessively noisy and potentially meaningless edges may be introduced, leading to performance degradation. So the value of k depends on $n_c$ to approximate.

### B IMPLETION DETAILS

For hyperparameters, we choose mask ratio $r = 0.2$, $\alpha_{mask} = 10$, $\alpha_{attn} = 0.1$, pertubation $\epsilon = 1e-5$. We use Adam as the optimizer with pretrain learning rate = 1e-3 and online learning rate encoder = 3e-3. The implementation details are shown in Table 8.

Table 8: Training and hyper-parameter settings of ConFVG. All experiments are run with a single NVIDIA 4090 GPU.

| Setting | Medical Insurance | YelpChi | Amazon |
|---|---|---|---|
| Optimizer | Adam | Adam | Adam |
| Learning rate (pretrain) | 1e-3 | 1e-3 | 1e-3 |
| Learning rate (online) | 3e-3 | 3e-3 | 3e-3 |
| Pre-training epochs | 100 | 100 | 100 |
| Number of pretrain tasks | 14 | 14 | 14 |
| Number of online tasks | 351 | 351 | 351 |
| Epochs per online task | 10 | 10 | 10 |
| Dropout | 0.5 | 0.5 | 0.5 |
| Feature mask ratio $r$ | 0.2 | 0.2 | 0.2 |
| $\alpha_{mask}$ | 10 | 10 | 10 |
| $\alpha_{attn}$ | 0.1 | 0.1 | 0.1 |
| Perturbation $\epsilon$ | 1e-5 | 1e-5 | 1e-5 |
| Top-$k$ components $n_c$ | 0.05 | 0.05 | 0.05 |
| EMA decay $\alpha$ | 0.99 | 0.99 | 0.99 |

Table 9: Comparison of scalability and complexity among different models. We compare the whole online learning time and trainable parameters with proposed models. CAREGNN, PCGNN, GTAN are static models without online learning period.

| Method | Online learning time (Minutes) | # Trainable Parameters |
|---|---|---|
| CAREGNN | / | 2.2K |
| PCGNN | / | 2.7K |
| GTAN | / | 61K |
| ContinuesGNN | 8.52 | 12K |
| POCL | 1.35 | 7.6K |
| **ConFVG (Ours)** | **5.46** | **11.8K** |

## C  COMPLEXITY AND SCALABILITY ANALYSIS

As shown in Table 9, we can see that our model has a relatively large number of parameters due to the autoencoder and mean-teacher architecture, yet it demonstrates better speed compared to ContinuesGNN during the online learning phase.

## D  SENSITIVITY ANALYSIS

Table 10: Hyperparameter sensitivity analysis.

| Value Metric | $\alpha_{mask}$ | | | | | | $\alpha_{attn}$ | | | | | |
|---|---|---|---|---|---|---|---|---|---|---|---|---|
| | 1 | 2 | 5 | 10 | 15 | 20 | 0.01 | 0.1 | 0.2 | 0.5 | 1 | 2 |
| AUC | 78.44 ± 0.60 | 79.48 ± 0.86 | 80.02 ± 0.90 | 80.48 ± 0.86 | 80.28 ± 0.94 | 80.02 ± 1.27 | 78.22 ± 0.72 | 80.48 ± 0.86 | 80.10 ± 0.76 | 79.33 ± 1.02 | 78.30 ± 1.22 | 76.22 ± 1.24 |
| F1-score | 62.36 ± 1.66 | 63.50 ± 1.57 | 64.12 ± 1.20 | 64.48 ± 1.24 | 64.28 ± 1.53 | 63.78 ± 1.44 | 64.25 ± 1.54 | 64.48 ± 1.24 | 64.01 ± 1.33 | 63.35 ± 1.62 | 62.28 ± 1.53 | 61.34 ± 1.82 |

| Value Metric | EMA $\alpha$ | | | | | | Perturbation $\varepsilon$ | | | | | |
|---|---|---|---|---|---|---|---|---|---|---|---|---|
| | 0.9 | 0.95 | 0.97 | 0.99 | 0.995 | 0.999 | 1e-6 | 1e-5 | 5e-5 | 1e-4 | 1e-3 | 1e-2 |
| AUC | 77.36 ± 1.55 | 79.35 ± 1.16 | 80.00 ± 1.03 | 80.48 ± 0.86 | 79.32 ± 1.34 | 78.82 ± 1.26 | 78.72 ± 0.53 | 80.48 ± 0.86 | 80.46 ± 0.78 | 80.33 ± 1.02 | 78.30 ± 1.22 | 76.05 ± 1.53 |
| F1-score | 61.46 ± 1.94 | 63.31 ± 1.37 | 64.48 ± 1.30 | 64.48 ± 1.24 | 64.17 ± 1.15 | 63.54 ± 1.72 | 64.02 ± 1.29 | 64.48 ± 1.24 | 64.52 ± 1.33 | 63.67 ± 1.67 | 63.08 ± 1.93 | 61.22 ± 1.86 |

We systematically evaluate the sensitivity of our model to key hyperparameters, with results presented in Table 10. The model exhibits strong robustness across a reasonably wide range of values. Performance consistently improves as the mask loss weight $\alpha_{mask}$ increases from 1 to 10, peaking at $\alpha_{mask} = 10$ with the highest AUC of 80.48 ± 0.86 and F1-score of 64.48 ± 1.24; further increasing $\alpha_{mask}$ beyond 10 leads to slight degradation and higher variance. The attention regularization coefficient $\alpha_{attn}$ achieves optimal results at 0.1, while values $\geq 0.5$ excessively suppress meaningful fraud-normal interaction patterns, causing a sharp performance drop. The EMA decay rate $\alpha$ performs best around 0.99. Perturbation parameter $\varepsilon$ shows excellent stability in the $[10^{-5}, 10^{-4}]$ range. Overall, the proposed approach maintains stable and superior performance across broad

hyperparameter intervals, fully demonstrating its robustness and practicality for real-world fraud detection deployment.

# E  ROBUSTNESS ANALYSIS

In addition to the scenarios we mentioned earlier, fraudsters may deliberately connect with normal users to conceal malicious activities, which would lead to significant attribute discrepancies between connected nodes. To evaluate our model robustness against adversarial fraud, we performed a controlled test by connecting 1%, 10%, 30%, 50% of fraud nodes to random normal nodes with 1 or 3 camouflage edges connected in medical insurance dataset with 10% label rate.

Table 11: Robustness of ConFVG with 1 or 3 camouflage edge connected.

| Camouflage edge | Proportion of fraud nodes | AUC | F1-score |
|---|---|---|---|
| | 0% (original graph) | 80.48 ± 0.86 | 64.48 ± 1.24 |
| | 1% | 80.12 ± 0.72 | 64.50 ± 1.52 |
| 1 | 10% | 79.39 ± 1.10 | 64.12 ± 1.48 |
| | 30% | 79.30 ± 1.35 | 63.29 ± 1.32 |
| | 50% | 78.70 ± 1.44 | 62.57 ± 0.71 |
| | 0% (original graph) | 80.48 ± 0.86 | 64.48 ± 1.24 |
| | 1% | 79.71 ± 0.77 | 65.10 ± 1.16 |
| 3 | 10% | 78.84 ± 1.02 | 63.70 ± 1.64 |
| | 30% | 78.82 ± 1.48 | 63.56 ± 1.57 |
| | 50% | 77.63 ± 1.65 | 63.14 ± 1.48 |

As shown in Table 11, ConFVG demonstrates remarkable robustness against deliberate camouflage attacks. When fraudsters randomly connect to only 1 normal node, even if 50% of fraud nodes perform such behavior, AUC drops merely from 80.48 to 78.70 ($\Delta = 1.78$). When the attack intensity is increased to 3 camouflage edges per fraud node, the performance decline is still gracefully controlled. These results indicate that the artificially injected camouflage edges, although intended to help fraudsters blend into normal communities, may inevitably create locally abnormal dense subgraphs or spectral perturbations. The Fiedler vector is highly sensitive to such structural irregularities and successfully captures them as useful guiding signals for community-aware learning, thereby suffering less from adversarial fraud.

# F  LLM USAGE

We utilized large language models (LLMs) to assist in drafting and refining this manuscript. Specifically, LLMs helped optimize language expression, enhance readability, and ensure clarity across sections. They assisted with tasks such as sentence restructuring, grammar checking, and improving the overall fluency of the text.

It should be noted that LLMs were not involved in the conceptualization, research design, or experimental planning. All research concepts, ideas, and analyses were independently developed and implemented by the authors. The contribution of LLMs was limited to enhancing the linguistic quality of the manuscript and did not involve scientific content or data analysis.

The authors take full responsibility for the integrity of the manuscript's content, including any text generated or refined by LLMs. We have ensured that the content produced by LLMs adheres to ethical standards and is free from plagiarism or academic misconduct.

