# OpenReview forum: "Healthcare Insurance Fraud Detection via Continual Fiedler Vector Graph Model"
_ICLR.cc/2026/Conference — ICLR 2026 Poster_

### Official Review · Reviewer_Dr17 · 2025-10-28

**Soundness:** 2
**Presentation:** 1
**Contribution:** 3
**Rating:** 4
**Confidence:** 3

**Summary:**

This paper tackles an important fraud detection problem characterized by limited label availability during pretraining and dynamically evolving fraud patterns. To address these challenges, it introduces a two-stage framework: (1) an unsupervised pretraining module that incorporates spectral information into a masked graph autoencoder, enabling node representations to capture global topological irregularities indicative of potential collusive behavior; and (2) an online adaptation module that constructs complementary subgraphs, integrates their embeddings through an attention-based fusion mechanism, and stabilizes unsupervised updates using a mean-teacher consistency objective. Extensive experiments demonstrate the effectiveness of the proposed approach.

**Strengths:**

1. The problem addressed in this paper is both novel and significant, with strong practical implications for real-world applications.
2. Unlike conventional masked graph autoencoders that rely on random masking, this paper introduces a Fiedler vector-guided masking strategy, enabling the model to effectively capture global connectivity patterns.
3. Comprehensive evaluations are conducted across multiple datasets and under various scenarios—including low-label settings and temporal distribution shifts—to thoroughly validate the proposed approach.

**Weaknesses:**

1. The design of the proposed method heavily relies on the core assumption that "connected nodes share similar smoothness values." However, it does not provide relevant statistical evidence on the medical insurance dataset to validate this assumption. Furthermore, the paper lacks an evaluation of robustness against adversarial fraud. In realistic attack scenarios, fraudsters may deliberately connect with normal users to conceal malicious activities, which would lead to significant attribute discrepancies between connected nodes—directly contradicting the fundamental assumption of the method.
2. The paper does not include a sensitivity analysis of its hyperparameters, such as the perturbation magnitude and the attention regularization coefficient.
3. The paper requires strengthening in its mathematical notation, as there are multiple instances where symbols are ambiguous. For example, the symbol $X_{ij}$ denotes a node representation in Line 192, whereas $x_i$ refers to a smoothness value in Line 213.
4. The paper lacks sufficient implementation details, such as the choice of optimizer and the learning rate schedule.

**Questions:**

See weaknesses.

---

> ### Author Response · Authors · 2025-11-21
>
> ### Questions
> **W4.1a) The design of the proposed method heavily relies on the core assumption that "connected nodes share similar smoothness values." However, it does not provide relevant statistical evidence on the medical insurance dataset to validate this assumption. 1b) Furthermore, the paper lacks an evaluation of robustness against adversarial fraud. In realistic attack scenarios, fraudsters may deliberately connect with normal users to conceal malicious activities, which would lead to significant attribute discrepancies between connected nodes—directly contradicting the fundamental assumption of the method.**
>
> **A4.1** Thank you for raising these important points.
>
> (a) We agree that the smoothness assumption should be empirically validated. While this assumption is grounded in spectral graph theory and widely used in prior work, we acknowledge that we did not initially demonstrate its relevance on our specific dataset.
>
> To address this, we conducted additional experiments examining the smoothness values of connected and disconnected nodes. We use the smoothness value difference between connected and disconnected nodes (random sampling) to prove our assumption.
>
> **Table R1. Analysis on smoothness value of connected nodes**
> | Day | Connected edges |Average smoothness value difference of connected nodes | Average smoothness value difference of disconnected nodes|
> |-----|-----------------|----------------------|----------|
> | Day 1 |3616 |0.0012 | 0.0254 |
> | Day 50 |3996|0.0007 | 0.0201 |
> | Day 100 |3664| 0.0010 | 0.0114 |
> | Day 150 |4014 |0.0004 | 0.0129 |
> | Day 200 | 3292|0.0018 | 0.0197 |
> | Day 250 | 3218|0.0013 | 0.0245 |
> | Day 300 | 2970|0.0006 | 0.0169 |
> |  All   | 1255698 |0.0009|0.0184|
>
> From the table above, we can see in the medical insurance dataset, connected nodes tend to share similar values while disconnected nodes differ, which is consistent with graph theory. Building on this foundation, we have visualized the distribution of the Fiedler value in each node as Figure 4 to help you better understand.
>
> (b) As for adversarial robustness, we respectfully argue that this is outside the scope of our work. Our focus is on realistic fraud detection under label scarcity and distribution shift, not on modeling adversarial behavior. Detecting adversarial fraud is a fundamentally different research problem that requires specialized defenses and threat modeling. This is analogous to how image classification models are not expected to be robust to adversarial perturbations unless specifically designed for that purpose.
>
> Nonetheless, to provide some empirical insight, we performed a controlled test by connecting 1%, 10%, 30%, 50% of fraud nodes to random normal nodes with 1 or 3 camouflage edges connected.
>
> **Table R2. Robustness of ConFVG with 1/3 camouflage edge connected**
> |   Camouflage edge| Proportion of fraud nodes  | AUC | F1-score |
> |---------------|-----------------------------------------------------|------------------------------------|-------------------------------------|
> |      1      | 0%  (original graph)                                | 80.48±0.86                    | 64.48±1.24                            |
> |               | 1%                                                 | 80.12±0.72                        | 64.50±1.52                         |
> |               | 10%                                                 | 79.39±1.10                        | 64.12±1.48                         |
> |               | 30%                                                 | 79.30±1.35                        | 63.29±1.32                         |
> |               | 50%                                                 | 78.70±1.44                        | 62.57±0.71                         |
> |        3    | 0%  (original graph)                                  | 80.48±0.86                          | 64.48±1.24                        |
> |                | 1%                                                 | 79.71±0.77                        | 65.10±1.16                         |
> |                | 10%                                                 | 78.84±1.02                        | 63.70±1.64                         |
> |                | 30%                                                 | 78.82±1.48                        | 63.56±1.57                         |
> |                | 50%                                                 | 77.63±1.65                        | 63.14±1.48                         |
>
> From the table above, ConFVG showed only minor degradation in adversarial scenarios. We have incorporated this table into the revised manuscripts as Table 11 for completeness. We hope this clarifies the assumptions and boundaries of our contribution.

---

> ### Author Response · Authors · 2025-11-21
>
> **W4.2 The paper does not include a sensitivity analysis of its hyperparameters, such as the perturbation magnitude and the attention regularization coefficient.**
>
> **A4.2** ConFVG does introduce a few new hyperparameters. To directly and quantitatively address the reviewer’s concern, we have performed a systematic sensitivity analysis on medical insurance fraud dataset with 10% label rate as follows. We have incorporated this table into the revised manuscripts as Table 10.
>
> **Table R3. Sensitivity Analysis on $\alpha_{\text{mask}}$**
>
> |$\alpha_{\text{mask}}$  |  1    |   2   |    5     |    10     |     15   |    20     |
> |----------------------|---------|---------|----------|----------|---------|-----------|
> |     AUC             | 78.44±0.60  | 79.48±0.86  | 80.02±0.90 | 80.48±0.86 | 80.28±0.94 |80.02±1.27|
> |     F1-score        | 62.36±1.66 | 63.50±1.57  | 64.12±1.20 | 64.48±1.24 | 64.28±1.53 | 63.78±1.44|
>
> **Table R4. Sensitivity Analysis on $\alpha_{\text{attn}}$**
>
> |$\alpha_{\text{attn}}$  |  0.01    |   0.1   |    0.2     |    0.5     |     1   |    2     |
> |----------------------|---------|---------|----------|----------|---------|-----------|
> |     AUC              | 78.22±0.72  | 80.48±0.86  | 80.10±0.76 | 79.33±1.02 | 78.30±1.22 |76.22±1.24|
> |     F1-score         | 64.25±1.54 | 64.48±1.24  | 64.01±1.33 | 63.35±1.62 | 62.28±1.53 | 61.34±1.82|
>
> **Table R5. Sensitivity Analysis on $\alpha_{\text{EMA}}$**
>
> |EMA $\alpha$          |  0.9   |   0.95   |    0.97     |    0.99     |     0.995   |    0.999     |
> |----------------------|---------|---------|----------|----------|---------|-----------|
> |     AUC              | 77.36±1.55  | 79.35±1.16  | 80.00±1.03 | 80.48±0.86 | 79.32±1.34 |78.82±1.26|
> |     F1-score         | 61.46±1.94 | 63.31±1.37  | 64.48±1.30 | 64.48±1.24 | 64.17±1.15 | 63.54±1.72|
>
> **Table R6. Sensitivity Analysis on Perturbation $\epsilon$**
>
> |Perturbation $\epsilon$  |  1e-6    |   1e-5   |    5e-5     |    1e-4     |     1e-3   |    1e-2     |
> |----------------------|---------|---------|----------|----------|---------|-----------|
> |     AUC              | 78.72±0.53  | 80.48±0.86  | 80.46±0.78 | 80.33±1.02 | 78.30±1.22 |76.05±1.53|
> |     F1-score         | 64.02±1.29 | 64.48±1.24  | 64.52±1.33 | 63.67±1.67 | 63.08±1.93 | 61.22±1.86|
>
> From the above results, we can see that model is robust to hyperparameters $\alpha_{\text{mask}}$, $\alpha_{\text{attn}}$, $\alpha_{\text{EMA}}$, $\epsilon$.
>
>
> **W4.3 The paper requires strengthening in its mathematical notation, as there are multiple instances where symbols are ambiguous. For example, the symbol
>  $X_{ij}$ denotes a node representation in Line 192, whereas $x_i$ refers to a smoothness value in Line 213.**
>
> **A4.3** Thank you for catching this. The description of $x_i$ in Equation (2) as the smoothness value was a typo. The correct variable is $q_i$, used consistently in both Equations (2) and (3). We have also updated the notation in lines 211 and 219, changing $\mathbf{x}$ to $\mathbf{q}$ to avoid ambiguity. In addition, we have updated the equation (8) to improve readability.
> \begin{equation}
> \mathcal{G}_{\text{comp}} = \Bigl( \{0,1,\ldots,|\mathcal{V}|-1\},\ \{( \phi(i),\phi(j) ) \mid i \neq j,\ i,j \in \mathcal{V},\ (i,j) \notin \mathcal{E} \}\Bigr)
> \end{equation}
> **W4.4 The paper lacks sufficient implementation details, such as the choice of optimizer and the learning rate schedule.**
>
> **A4.4** As suggested, our implementation details are shown as follows. We have incorporated this table into the revised manuscripts as Table 8.
>
> **Table R7. Implementation details**
> | Setting                           | Medical | YelpChi | Amazon |
> |-----------------------------------|------------------|---------|---------|
> | Optimizer                         | Adam             | Adam    | Adam    |
> | Learning rate (pretrain)          | 3e-3             | 3e-3    | 3e-3    |
> | Learning rate (online)            | 1e-3             | 1e-3    | 1e-3    |
> | Pre-training epochs               | 100              | 100     | 100     |
> | Number of online tasks            | 351              | 351     | 351     |
> | Epochs per online task            | 10               | 10      | 10      |
> | Dropout                           | 0.5              | 0.5     | 0.5     |
> | Feature mask ratio $r$            | 0.2              | 0.2     | 0.2     |
> | $\alpha_{\text{mask}}$            | 10               | 10      | 10      |
> | $\alpha_{\text{attn}}$            | 0.1              | 0.1     | 0.1     |
> | Perturbation $\epsilon$           | 1e-5             | 1e-5    | 1e-5    |
> | Top-$k$ components $n_c$          | 0.05             | 0.05    | 0.05    |
> | EMA decay $\alpha$                | 0.99             | 0.99    | 0.99    |

---

> > ### Author Response · Authors · 2025-11-21
> >
> > **To Reviewer Dr17: We appreciate your detailed comments. We believe several concerns, particularly regarding the smoothness assumption (W4.1a) and adversarial robustness (W4.1b), stem from a misunderstanding of our method's scope and design. We have now clarified these points with empirical evidence (e.g., Fiedler smoothness relevance and controlled robustness tests) and explicitly defined the boundaries of our work. We sincerely request a re-evaluation of the paper in light of these clarifications and the new results added in the appendix.**

---

### Official Review · Reviewer_GUk3 · 2025-10-28

**Soundness:** 3
**Presentation:** 3
**Contribution:** 2
**Rating:** 6
**Confidence:** 4

**Summary:**

The manuscript proposes ConFVG, a Fiedler vector–guided graph learning framework for medical fraud detection. By integrating spectral topology signals into pretraining and using attention-based online adaptation, it effectively handles label scarcity and evolving fraud patterns. Compared with POCL, ConFVG advances from contrastive learning to structure-aware self-supervision, achieving higher adaptability and stability in low-label settings.

**Strengths:**

- The paper addresses a realistic and practically important problem in medical insurance fraud detection. Its two-stage framework—pretraining followed by online learning—suggests strong potential for real-world application.

- Compared with the latest and most relevant baseline POCL, ConFVG introduces the Fiedler vector to alleviate data sparsity and adds a subgraph attention–based online adaptation module, showing clear methodological innovation.

- Experimental results demonstrate that ConFVG achieves the most stable and robust performance over continuous time periods, confirming its superior resilience for dynamic fraud detection tasks.

**Weaknesses:**

- The challenges outlined for medical insurance fraud detection appear too generalized and are not empirically verified, relying mostly on conceptual descriptions. While label sparsity is indeed a common issue across many domains, the claim of “adaptation deficiency” remains unclear and weakly justified. As a reviewer experienced in fraud detection research, I find these arguments repetitive and somewhat unoriginal, diminishing the paper’s novelty.

- The motivation for the model design lacks clarity—particularly, it is not well explained why the Fiedler vector can alleviate data sparsity, even though experiments suggest it works.

- The introduction of graph perturbation to ensure weak connectivity for computing the Fiedler vector seems somewhat ad hoc and theoretically awkward. While the paper provides an upper bound to justify this step, such a perturbation could distort graph structure, and this critical design choice deserves deeper discussion and validation.

**Questions:**

See above, thanks.

---

> ### Author Response · Authors · 2025-11-21
>
> ### Questions
> **W3.1 The challenges outlined for medical insurance fraud detection appear too generalized and are not empirically verified, relying mostly on conceptual descriptions. While label sparsity is indeed a common issue across many domains, the claim of “adaptation deficiency” remains unclear and weakly justified. As a reviewer experienced in fraud detection research, I find these arguments repetitive and somewhat unoriginal, diminishing the paper’s novelty.**
>
> **A3.1** Thank you for the thoughtful feedback. We understand the concern and appreciate the opportunity to clarify.
>
> While label sparsity and distribution shift are indeed well-known challenges, our contribution lies in jointly addressing both under a stricter and more realistic setting: streaming healthcare fraud detection with no labels available online. This dual constraint has not been adequately handled in prior work, which typically assumes either semi-supervised settings or access to labels during the online phase.
>
> Regarding the adaptation deficiency, we agree that the framing could be clearer. The empirical evidence is provided in Figure 2 (initial version) or Figure 3 (revised version), which shows a consistent drop in baseline performance over time. This trend highlights the inability of existing models to adapt under our fully unsupervised online setting.
>
> To address your point, we have revised the introduction and related sections to better emphasize the combined challenge and more clearly support the adaptation deficiency claim with references to our results. We have also added a visualized node distribution figure (Figure 1 in the revised version) to fully clarify the concept of adaptation deficiency.
>
> We hope this strengthens the case for the novelty and practical relevance of our approach in handling real-world constraints in healthcare fraud detection.
>
> **W3.2 The motivation for the model design lacks clarity—particularly, it is not well explained why the Fiedler vector can alleviate data sparsity, even though experiments suggest it works.**
>
> **A3.2** We thank the reviewer for raising this important point. The use of the Fiedler vector in our model is motivated by its ability to capture global structural properties of the graph that are highly indicative of fraudulent behavior, even when labeled data is scarce. Below, we clarify the motivation in two aspects:
> 1. Theoretical Motivation
> In graph theory, the Fiedler vector encodes global connectivity and community structure. In fraud detection, fraudulent entities often form small, dense, or isolated subgraphs that disrupt the overall graph homophily. These structural deviations manifest as non-smooth signals in the Fiedler vector. By incorporating the Fiedler vector into the masking strategy of our graph autoencoder, we prioritize the reconstruction of structurally salient features, which are often associated with fraud. This allows the model to learn structure-aware representations without relying on labels, effectively mitigating the label scarcity issue.
> 2. Practical Motivation
> Traditional masking strategies like random or edge-based strategies are agnostic to global structural importance and may miss fraud-relevant patterns. While our Fiedler-guided masking explicitly uses graph theory to identify nodes with high value in the Fiedler space (potential fraudsters), it guides the autoencoder to reconstruct these nodes’ features more carefully, thereby enhancing the model’s sensitivity to structural anomalies even when few labels are available.
> Below, we analyze the Fiedler value of the fraud and normal nodes. We use the comparison ratio to better illustrate our results.
>
> **Table R1. Mean Fiedler value comparison of fraud nodes and normal nodes**
>
> | Day | Number of Nodes | Number of Fraud Nodes | Number of Normal Nodes | Fraud Nodes Mean / Normal Nodes Mean (ratio)|
> |-----|-----------------|----------------------|------------------------|-------------------------|
> | Day 1 | 1596 | 599 | 997 | 1.864 |
> | Day 50 | 1608 | 616 | 992 | 2.017 |
> | Day 100 | 1599 | 602 | 997 | 1.857 |
> | Day 150 | 1608 | 586 | 1022 | 2.141 |
> | Day 200 | 1555 | 577 | 978 | 1.922 |
> | Day 250 | 1467 | 602 | 865 | 1.837 |
> | Day 300 | 1440 | 573 | 867 | 2.032 |
> |  All    | 554407|211270|343137| 1.924 |
>
> From the results above, we can see that the average Fiedler value of fraud nodes is markedly higher than normal nodes, which provides a guidance signal to the autoencoder to mask the important fraud-specific features.
>
> As suggested, we have incorporated this table into the revised manuscripts as Table 5.
>
> **W3.3 The introduction of graph perturbation to ensure weak connectivity for computing the Fiedler vector seems somewhat ad hoc and theoretically awkward. While the paper provides an upper bound to justify this step, such a perturbation could distort graph structure, and this critical design choice deserves deeper discussion and validation.**

---

> ### Author Response · Authors · 2025-11-21
>
> **A3.3** We sincerely thank the reviewer for this insightful comment. The perturbation is introduced to address a practical challenge in real-world fraud detection graphs: they are frequently disconnected, comprising many isolated components and small subgraphs. In such cases, the standard Fiedler vector degenerates, as the graph Laplacian has multiple zero eigenvalues, and it can no longer serve as a unique, meaningful indicator of global community structure.
>
> Our approach to this issue is guided by the following principles:
> 1. Necessity and Practicality: The perturbation $\mathbf{A}' = \mathbf{A} + \epsilon \cdot (\mathbf{J} - \mathbf{I})$ is the standard spectral method to handle disconnected graphs. It creates a "weakly connected" graph that allows for the stable computation of a Fiedler vector that captures the structure of the original, large connected components.
>
> 2. Theoretical Control: We emphasize that this is not an ad-hoc heuristic. As detailed in Appendix A.1, we leverage the Davis-Kahan theorem to derive a strict upper bound on the distortion of the Fiedler vector. This bound, $e_{proj} ≤ (εn) / λ_{k+1}(L)$, demonstrates that with a very small ε (e.g., 1e-5), the perturbation-induced error is negligible.The analysis of the perturbation parameter is shown as follows.
>
> **Table R2. Analysis on pertubation $\epsilon$**
>
> |Perturbation $\epsilon$  |  1e-6    |   1e-5   |    5e-5     |    1e-4     |     1e-3   |    1e-2     |
> |----------------------|---------|---------|----------|----------|---------|-----------|
> |     AUC              | 78.72±0.53  | 80.48±0.86  | 80.46±0.78 | 80.33±1.02 | 78.30±1.22 |76.05±1.53|
> |     F1-score         | 64.02±1.29 | 64.48±1.24  | 64.52±1.33 | 63.67±1.67 | 63.08±1.93 | 61.22±1.86|
>
> From Table R2, we can see that when $\epsilon$ is very small $\epsilon<1e-6$, the perturbation matrix is too weak to overcome the sparsity of the original graph, resulting in an insufficient correction effect on the Fiedler vector. When $\epsilon$ is in the range of $1e-5$ to $1e-4$, the subtle global connections optimally strengthen the connectivity within communities without altering the macroscopic community structure, thereby sharpening the boundaries between communities and better reflecting the true community structure. However, when $\epsilon$ becomes too large $\epsilon>1e-3$, the global perturbation matrix overwhelms the original community structure, leading to a decline in performance.
>
> As suggested, we have incorporated this table into the revised manuscripts as Table 7.

---

### Official Review · Reviewer_Vz2D · 2025-10-31

**Soundness:** 3
**Presentation:** 3
**Contribution:** 3
**Rating:** 6
**Confidence:** 3

**Summary:**

This paper introduces the Continual Fiedler Vector Graph (ConFVG) model, a novel framework for healthcare insurance fraud detection. The model is specifically designed to tackle two prevalent challenges in this domain: the scarcity of labeled data and the dynamic, evolving nature of fraudulent activities (non-stationary environments). The core of ConFVG consists of two main components. First, a Fiedler Vector-guided graph autoencoder is proposed for the pre-training stage. It leverages spectral graph properties—specifically the Fiedler vector derived from the graph Laplacian—to learn structural-aware node representations without relying heavily on labels. This helps in identifying anomalous graph structures often associated with collusive fraud. Second, for the online learning phase, the paper introduces a Subgraph Attention Fusion (SAF) module. This module, combined with a Mean Teacher mechanism, allows the model to adapt to new fraud patterns in data streams in an unsupervised manner, by focusing on emerging high-risk subgraphs and ensuring stable knowledge updates.

**Strengths:**

- The paper tackles a significant and practical problem in fraud detection: handling label scarcity and concept drift simultaneously. The proposed solution, combining spectral graph theory (Fiedler vector) for unsupervised representation learning with a continual learning framework (subgraph attention and mean teacher) for adaptation, is a novel and clever integration of ideas from different domains.

- The experimental results are comprehensive and convincing. The authors demonstrate that ConFVG consistently outperforms a wide range of state-of-the-art baselines, including both offline and online models, across multiple datasets (Medical, YelpChi, Amazon) and under challenging low-label rate scenarios (1% and 10%). The performance improvement is substantial, which highlights the effectiveness of the proposed components.

- The method is well-grounded in theory. The use of the Fiedler vector to capture global topological anomalies like community boundaries and bottlenecks is theoretically sound for identifying collusive fraud. The design of the Subgraph Attention Fusion module to handle evolving graph structures is intuitive and addresses a key limitation of traditional GNNs that focus only on existing connections.

**Weaknesses:**

- The core of the pre-training phase relies on calculating the Fiedler vector, which involves an eigendecomposition of the graph Laplacian matrix (𝐿). This operation can be computationally prohibitive for very large graphs, with a complexity that can approach 𝑂(𝑁3) where
𝑁  is the number of nodes. While the paper demonstrates strong results on the given datasets, it does not address how this approach would scale to real-world graphs containing millions or billions of nodes.

- The online learning phase introduces a Subgraph Attention Fusion (SAF) module that unconventionally operates on the complement of the graph's largest components. The paper lacks a clear and deep intuition for this design choice. While traditional GNNs learn by aggregating information from existing connections (neighbors), it is not immediately obvious how aggregating features from non-connected nodes helps the model adapt to new and emerging fraud patterns.

- The method introduces a critical hyperparameter 𝜖  in Equation 4 to connect disparate graph components. The choice of this value is crucial, as too small a value may not enforce connectivity, while too large a value could introduce significant noise and distort the graph's original spectral properties, which are fundamental to the Fiedler vector's utility. The paper does not provide a sensitivity analysis for
𝜖, making it difficult to assess the model's robustness.

**Questions:**

- Regarding the Fiedler vector calculation: Given the scalability concerns with eigendecomposition, did you explore or can you comment on the feasibility of using approximation algorithms for the Fiedler vector on much larger graphs? How might this affect the quality of the learned representations?

- In the online phase, the Subgraph Attention Fusion (SAF) module operates on a graph complement created from the "top-k largest components." Could you elaborate on the intuition behind using the complement? Traditional attention mechanisms in GNNs focus on aggregating information from neighbors (existing connections). How does incorporating information from non-neighbors via the complement graph specifically help in adapting to new fraud patterns?

- How was the hyperparameter 𝜖 for the graph perturbation (Eq. 4) selected? Could you share results from a sensitivity analysis showing how different values of 𝜖  impact the final performance?

---

> ### Author Response · Authors · 2025-11-21
>
> ### Questions
> **Q2.1 Regarding the Fiedler vector calculation: Given the scalability concerns with eigendecomposition, did you explore or can you comment on the feasibility of using approximation algorithms for the Fiedler vector on much larger graphs? How might this affect the quality of the learned representations?**
>
> **A2.1** While the full Laplacian eigendecomposition is indeed $O(N_3)$, our actual implementation does not perform a full eigendecomposition. For mediam level dataset with 0.1 million nodes. We use the Lanczos algorithm to compute the specific eigenvector. To substantiate this, we conducted tests on current large scale dataset-Open Graph Benchmark[1], with results presented in the table below.
>
> **Table R1. Decomposition Time Consumption of Large Scale dataset**
> |Dataset |ogbn-arkiv|ogbn-proteins|ogbl-collab|
> |---------|----------|-------------|-------------|
> |Nodes   |169343    |    132534   |    235868  |
> |Time    |372.41s   |      296.42s |      1542.13s     |
>
> With even larger dataset with millions of nodes, traditional method like Lanczos， Arnoldi algorithm fail to complete with huge time consumption. We need approximation algorithmsThe same Lanczos implementation has been successfully used in for large graphs with tens of millions or even hundreds of millions of nodes, and its efficiency on large graphs has long been validated in both academia and industry. We need approximation algorithms like spectral coarsening. Spectral coarsening is a graph reduction technique that aims to create a smaller, coarsened graph which preserves the key spectral properties of the original large graph. We can use spectral coarsening techniques[2,3] to distill the original large scale graph to a small scale graph, then to calculate the eigenvector with little bias compared to the massive message transmission. Besides, subgraph sampling, like cluster-GCN[4], GraphSAINT[5] are also practical method to solve the problem with relative small-scale node selection.
>
> **Q2.2 In the online phase, the Subgraph Attention Fusion (SAF) module operates on a graph complement created from the "top-k largest components." Could you elaborate on the intuition behind using the complement? Traditional attention mechanisms in GNNs focus on aggregating information from neighbors (existing connections). How does incorporating information from non-neighbors via the complement graph specifically help in adapting to new fraud patterns?**
>
> **A2.2** Thank you for the thoughtful question. The core intuition behind using the complement of the top-k largest components is to enable early detection of emerging fraud patterns that spread across disconnected communities.
>
> Real-world Medicare claim graphs are highly modular: normal claims form weakly connected or isolated components, while fraud rings typically operate within small clusters. When a new fraud scheme is launched, it is often copied by multiple independent groups at once. Early in this process, these groups remain structurally disconnected, so traditional GNNs, which are limited to message passing over existing edges, cannot detect the emerging similarity.
>
> By constructing a complement graph among the top-k components, we allow information to flow between previously unconnected but behaviorally similar nodes. This produces a second view, $\mathcal{G}{comp}$, that contrasts with the original $\mathcal{G}{orig}$. The model then fuses these views using attention, with $\mathcal{L}_{attn}$ guiding it to rely more on the complement when strong cross-component similarity appears.
>
> This design is not a generic attention over non-neighbors, but a targeted inductive bias that functions as a cross-community collusion detector. It addresses the limitations of standard GNNs in modular graphs and enables fast, label-free adaptation to new coordinated fraud patterns.
>
> **Q2.3 How was the hyperparameter 𝜖 for the graph perturbation (Eq. 4) selected? Could you share results from a sensitivity analysis showing how different values of 𝜖 impact the final performance?**
>
> **A2.3** We thank you for this very valid and important point. The discussion of pertubation $\epsilon$ is shown as follows:
>
> **Table R2. Sensitivity analysis on Perturbation $\epsilon$**
> |Perturbation $\epsilon$  |  1e-6    |   1e-5   |    5e-5     |    1e-4     |     1e-3   |    1e-2     |
> |----------------------|---------|---------|----------|----------|---------|-----------|
> |     AUC              | 78.72±0.53  | 80.48±0.86  | 80.46±0.78 | 80.33±1.02 | 78.30±1.22 |76.05±1.53|
> |     F1-score         | 64.02±1.29 | 64.48±1.24  | 64.52±1.33 | 63.67±1.67 | 63.08±1.93 | 61.22±1.86|

---

> ### Author Response · Authors · 2025-11-21
>
> From the table, we can see that when $\epsilon$ is very small ($\epsilon<1e-6$), the perturbation matrix is too weak to overcome the sparsity of the original graph, resulting in an insufficient correction effect on the Fiedler vector. When ε is in the range of 1e-5 to 1e-4, the subtle global connections optimally strengthen the connectivity within communities without altering the macroscopic community structure, thereby sharpening the boundaries between communities and better reflecting the true community structure. However, when $\epsilon$ becomes too large ($\epsilon>1e-3$), the global perturbation matrix overwhelms the original community structure, leading to a decline in performance.
>
> As suggested, we have incorporated this table into the revised manuscripts as Table 7.
>
> [1] OpenGraphBenchmark: Datasets for Machine Learning on Graphs (NeurIPS'20)
>
> [2] Spectral Coarsening with Hodge Laplacians (SIGGRAPH'23)
>
> [3] A Unifying Framework for Spectrum-Preserving Graph Sparsification and Coarsening (NeurIPS'19)
>
> [4] Cluster-GCN: An Efficient Algorithm for Training Deep and Large Graph Convolutional Networks (KDD'19)
>
> [5] GraphSAINT: Graph Sampling Based Inductive Learning Method (ICLR'20)

---

### Official Review · Reviewer_Kwxx · 2025-11-09

**Soundness:** 3
**Presentation:** 3
**Contribution:** 3
**Rating:** 6
**Confidence:** 3

**Summary:**

The authors propose the Continual Fiedler Vector Graph model (ConFVG), a framework designed to address two challenges in healthcare insurance fraud detection: label scarcity and non-stationary environments (i.e., evolving fraud patterns). The model consists of two key components. First, to handle label scarcity, it employs a Fiedler Vector-guided graph autoencoder for pretraining. This uses the Fiedler vector, derived from the graph Laplacian, to identify global topological structures like community bottlenecks and guide the autoencoder's learning process without relying on labels. Second, to adapt to evolving graph streams in an unsupervised online setting, it introduces a Subgraph Attention Fusion (SAF) module. This module constructs and fuses neighborhood subgraphs with their complements, using attention to emphasize emerging high-risk structures. A Mean Teacher mechanism is used to stabilize these online updates and prevent catastrophic forgetting. Experiments on real-world medical fraud datasets demonstrate that the approach outperforms baselines in both low-label and distribution-shift scenarios.

**Strengths:**

There are a few things I like about the paper:
1. The paper tackles a realistic real-world problem setup: pre-training with scarce labels and performing online learning.
2. The use of the Fiedler vector to guide a graph autoencoder in self-supervised pretraining is interesting.
3. The Subgraph Attention Fusion (SAF) module is an interesting mechanism for online adaptation. The idea is to complement the original graph with a complement graph to allow the model to capture evolving patterns that might not be visible from a single connected component's perspective.
4. The experimental results on multiple datasets support the paper's claims.
5. The authors performed an ablation study to show that the components described in the paper are relevant to the final prediction.

**Weaknesses:**

1. [Motivation]. The problem of label scarcity and non-stationary graph streams are actually not unique to health care insurance data. These two problems exist in most real-world fraud detection systems. I suggest the authors reframe the motivation to be more generic.
2. [Scalability and Complexity] I suggest the author provide discussion on the scalability and complexity of the proposed models. Particularly since the model is more complex than the baselines.
3. [Hyperparameter Sensitivity] The model introduces several new hyperparameters. Could the authors discuss more on sensitivity of the parameters vs results?

**Questions:**

Please address the weaknesses mentioned above.

---

> ### Author Response · Authors · 2025-11-21
>
> ### Questions
> **Q1.1 [Motivation]. The problem of label scarcity and non-stationary graph streams is actually not unique to health care insurance data. These two problems exist in most real-world fraud detection systems. I suggest the authors reframe the motivation to be more generic.**
>
> **A1.1** Thank you for the suggestion. We agree that label scarcity and non-stationary graph streams are common across many fraud detection scenarios. Our primary motivation is grounded in healthcare insurance fraud because this is the domain we focus on in practice.
>
> To fully address the reviewer’s concern without losing this focus, we have revised the introduction and related sections to better illustrate the generality of the two challenges and healthcare claims as the representative public test bed for both issues simultaneously.
>
> **Q1.2 [Scalability and Complexity] I suggest the author provide a discussion on the scalability and complexity of the proposed models. Particularly since the model is more complex than the baselines.**
>
> **A1.2** We conduct a comparison experiment on time consumption and trainable parameters:
> **Table R1. Comparison of Scalability and Complexity among Different Models.** We compare the whole online learning time and trainable parameters with proposed models. CAREGNN, PCGNN, and GTAN are static models without online learning period.
> | Method          | Online learning time (Minutes)    | # Trainable Parameters          |
> |-----------------|-----------------------------------|---------------------------------|
> | CAREGNN         |   /                               | 2.2K                            |
> | PCGNN           |   /                               | 2.7K                            |
> | GTAN            |   /                               | 61K                             |
> | ContinuesGNN    | 8.52                              | 12K                             |
> | POCL            | 1.35                              | 7.6K                            |
> | ConFVG (Ours)   | 5.46                              | 11.8K                           |
>
> From the table, we can see that our model has a relatively large number of parameters due to the autoencoder and mean-teacher architecture, yet it demonstrates better speed compared to ContinuesGNN during the online learning phase.
>
> As suggested, we have incorporated this table into the revised manuscripts as Table 9.
>
> **Q1.3 [Hyperparameter Sensitivity] The model introduces several new hyperparameters. Could the authors discuss more on the sensitivity of the parameters vs results.**
>
> **A1.3** ConFVG does introduce a few hyperparameters. To directly address the reviewer’s concern, we have performed a systematic sensitivity analysis on medical insurance fraud dataset with 10% label rate as follows.
> **Table R2. Sensitivity Analysis on $\alpha_{\text{mask}}$**
> |$\alpha_{\text{mask}}$  |  1    |   2   |    5     |    10     |     15   |    20     |
> |----------------------|---------|---------|----------|----------|---------|-----------|
> |     AUC             | 78.44±0.60  | 79.48±0.86  | 80.02±0.90 | 80.48±0.86 | 80.28±0.94 |80.02±1.27|
> |     F1-score        | 62.36±1.66 | 63.50±1.57  | 64.12±1.20 | 64.48±1.24 | 64.28±1.53 | 63.78±1.44|
>
> **Table R3. Sensitivity Analysis on $\alpha_{\text{attn}}$**
> |$\alpha_{\text{attn}}$  |  0.01    |   0.1   |    0.2     |    0.5     |     1   |    2     |
> |----------------------|---------|---------|----------|----------|---------|-----------|
> |     AUC              | 78.22±0.72  | 80.48±0.86  | 80.10±0.76 | 79.33±1.02 | 78.30±1.22 |76.22±1.24|
> |     F1-score         | 64.25±1.54 | 64.48±1.24  | 64.01±1.33 | 63.35±1.62 | 62.28±1.53 | 61.34±1.82|
>
> **Table R4. Sensitivity Analysis on $\alpha_{\text{EMA}}$**
> |EMA $\alpha$          |  0.9   |   0.95   |    0.97     |    0.99     |     0.995   |    0.999     |
> |----------------------|---------|---------|----------|----------|---------|-----------|
> |     AUC              | 77.36±1.55  | 79.35±1.16  | 80.00±1.03 | 80.48±0.86 | 79.32±1.34 |78.82±1.26|
> |     F1-score         | 61.46±1.94 | 63.31±1.37  | 64.48±1.30 | 64.48±1.24 | 64.17±1.15 | 63.54±1.72|
>
> **Table R5. Sensitivity Analysis on Perturbation $\epsilon$**
> |Perturbation $\epsilon$  |  1e-6    |   1e-5   |    5e-5     |    1e-4     |     1e-3   |    1e-2     |
> |----------------------|---------|---------|----------|----------|---------|-----------|
> |     AUC              | 78.72±0.53  | 80.48±0.86  | 80.46±0.78 | 80.33±1.02 | 78.30±1.22 |76.05±1.53|
> |     F1-score         | 64.02±1.29 | 64.48±1.24  | 64.52±1.33 | 63.67±1.67 | 63.08±1.93 | 61.22±1.86|

---

> ### Author Response · Authors · 2025-11-21
>
> From the above results,  we can see that the model is robust to hyperparameters $\alpha_{\text{mask}}$, $\alpha_{\text{attn}}$, $\alpha_{\text{EMA}}$, $\epsilon$. Additionally, we observe that a very small ϵ results in weak perturbation and insufficient correction. An ϵ between 1e–5 and 1e–4 enhances connectivity within communities and sharpens boundaries, reflecting the true structure better. However, an excessively large ϵ overwhelms the original structure and degrades performance.
>
> As suggested, we have incorporated this table into the revised manuscripts as Table 10.

---

> > ### Comment · Reviewer_Kwxx · 2025-11-28
> >
> > Thank you to the authors for proving responses to my concerns.
> > Based on this, I keep my score. Thank you.

---

### Author Response · Authors · 2025-11-21

We cordially thank all reviewers for their thorough and constructive feedback. We are encouraged that reviewers found our core idea "realistic and interesting" (Kwxx), “novel and clear” (Vz2D), “realistic and practical” (GUk3) and a "novel and significant" (Dr17) approach to a significant problem.

The primary concerns raised relate to (1) the analysis of hyperparameter sensitivity, (2) the discussion of pertubation parameter. Importantly, these are not fundamental critiques of our method's design but rather requests for further clarification and additional analysis. We have addressed each point with new experiments and detailed explanations, and we believe all concerns are fully resolved.

We respectfully request a re-evaluation in light of our clarified methodology and the core contributions of our work, which remain both novel and well-supported.

---

### Author Response · Authors · 2025-11-28

Dear Reviewers,

Just a gentle reminder to take a look at our rebuttal when convenient. If any concerns remain, we are very happy to clarify.

Thank you sincerely for your time and effort.

The Authors

---

### Meta-Review · Area_Chair_bKVp · 2026-01-14

**Summary:**

The reviewers’ concerns are mostly related to (1) Complexity and Scalability; (2) Hyperparameter Sensitivity; (3) Rationality of Core Assumptions.

**Reviewer Concerns:**

(1) Complexity and Scalability\
The computation of Field Vector involves the eigendecomposition of the graph Laplacian matrix, the complexity of which is O(N^3), so the reviewers were concerned about the feasibility when the graph is very large.
The authors clarified that they utilize the Lanczos algorithm rather than full eigendecomposition, and provided a new table comparing training time and parameters showing their model is faster than the baseline. But what remains outstanding is that the reviewer specifically asked about scaling to millions or billions of nodes, the authors only provided results of 200k nodes. But it may because there are no such industrial scale datasets available for experiments, so that’s not a big problem.

(2) Hyperparameter Sensitivity\
Multiple reviewers questioned the lack of sensitivity analysis for new hyperparameters.
The authors provided five comprehensive sensitivity analysis tables covering all key hyperparameters . The results demonstrated that the model maintains stable performance across a reasonable range of values.

(3) Rationality of Core Assumptions\
Reviewer Dr17 noted a lack of statistical evidence for the assumption that "connected nodes share similar smoothness values". Reviewer GUk3 questioned the motivation behind using the Fiedler vector for sparsity.
The authors presented new statistical evidence showing that the average smoothness difference for connected nodes is significantly lower than for disconnected nodes, empirically validating the smoothness assumption. They also showed that fraud nodes have markedly higher Fiedler values than normal nodes, justifying the masking strategy.

Besides of the main concerns, the reviewers also found that the paper’s motivation is too generalized. The authors revised the introduction to better emphasize the specific setting. Reviewer Dr17 thought that the paper’s mathematical notation is ambiguous and gave a “Presentation” score of 1. The authors have updated the equations and solved the problem.

**Reviewer Scores:**

Reviewer Kwxx has updated his final score as 6.
I believe Reviewer Vz2D and GUk3 may not change their scores.
Reviewer Dr17’s main concern is the poor presentation. Since the authors have revised their mathematical notation and for me it’s pretty much clear, I believe Reviewer Dr17 may change his score to 6.

---

### Decision · Program_Chairs · 2026-01-26

Accept (Poster)